# Statistical and Topological Properties of Sliced Probability Divergences

**Kimia Nadjahi**[1]*,     **Alain Durmus**[2],     **Lénaïc Chizat**[3],
**Soheil Kolouri**[4],     **Shahin Shahrampour**[5],     **Umut Şimşekli**[1,6]

1: LTCI, Télécom Paris, Institut Polytechnique de Paris, France
2: Centre Borelli, ENS Paris-Saclay, CNRS, Université Paris-Saclay, France
3: Laboratoire de Mathématiques d'Orsay, CNRS, Université Paris-Saclay, France
4: HRL Laboratories, LLC., Malibu, CA, USA
5: Texas A&M University, College Station, TX, USA
6: Department of Statistics, University of Oxford, UK

## Abstract

The idea of slicing divergences has been proven to be successful when comparing two probability measures in various machine learning applications including generative modeling, and consists in computing the expected value of a 'base divergence' between *one-dimensional random projections* of the two measures. However, the topological, statistical, and computational consequences of this technique have not yet been well-established. In this paper, we aim at bridging this gap and derive various theoretical properties of sliced probability divergences. First, we show that slicing preserves the metric axioms and the weak continuity of the divergence, implying that the sliced divergence will share similar topological properties. We then precise the results in the case where the base divergence belongs to the class of integral probability metrics. On the other hand, we establish that, under mild conditions, the sample complexity of a sliced divergence does not depend on the problem dimension. We finally apply our general results to several base divergences, and illustrate our theory on both synthetic and real data experiments.

## 1   Introduction

Most inference methods in implicit generative modeling (IGM), such as generative adversarial networks [1] and variational auto-encoders [2], rely on the use of a particular divergence in order to be able to discriminate probability distributions. Recent advances in this field have illustrated that the choice of this divergence is of crucial importance and can lead to very different practical and theoretical properties [3, 4, 5, 6, 7]. In this context, 'sliced' probability divergences, such as Sliced-Wasserstein [8], or Sliced-Cramér [9], have become increasingly popular.

This slicing strategy has been essentially motivated by two main purposes. The first purpose is that some probability divergences are only defined to compare measures supported on one-dimensional spaces (e.g., Cramér distance, [10]); hence, the slicing operation allows the use of such divergences to multivariate distributions [11, 9]. The second purpose arises when the computational complexity of a divergence becomes excessive when comparing measures on high-dimensional spaces, but can efficiently be computed in the univariate case (e.g., the Wasserstein distance between one-dimensional distributions admits a closed-form analytical which can easily be approximated). The slicing operation then leverages these advantages originally available in one dimension to define divergences achieving computational efficiency on multivariate settings [8, 12, 13, 14, 15, 16].

Even though various sliced divergences have successfully been deployed in practical applications, their theoretical properties have not yet been well-understood. Existing results are largely restricted to the specific case of the Sliced-Wasserstein (SW) distance: it has been shown that SW satisfies the metric axioms [17], the convergence in SW is equivalent to the convergence in Wasserstein distance [18, 19], and the estimators obtained by minimizing SW are consistent [18]. Besides, some properties of SW have only been characterized for specific settings, in particular its statistical benefits observed in practice [20, 12]. In this paper, we aim to bridge this gap by investigating the theoretical properties of sliced probability divergences from a general point of view: since such divergences are all characterized via the same slicing operation, we explore in depth the topological and statistical implications of this operation. Specifically, we consider a generic base divergence $\mathbf{\Delta}$ between one-dimensional probability measures, and define its sliced version, denoted by $\mathbf{S\Delta}$, which operates on multivariate settings.

We first establish several topological properties of $\mathbf{S\Delta}$. Thanks to our general approach, our findings can directly be applied to any instance of sliced divergence, including those motivated by the two aforementioned purposes. Specifically, we show that slicing preserves the metric properties: if $\mathbf{\Delta}$ is a metric, so is $\mathbf{S\Delta}$ (Proposition 1). We then focus on finer topological properties of $\mathbf{S\Delta}$ and show in Theorem 1 that, if the convergence in $\mathbf{\Delta}$ implies the weak convergence of measures (or conversely), then slicing preserves this property, *i.e.* the convergence in $\mathbf{S\Delta}$ implies the weak convergence of measures (or conversely). We also consider the case when $\mathbf{\Delta}$ is an integral probability metric [21] and identify sufficient conditions for $\mathbf{S\Delta}$ to be upper-bounded by $\mathbf{\Delta}$, which implies that $\mathbf{S\Delta}$ induces a weaker topology (Theorem 2). Similarly, we identify sufficient conditions such that $\mathbf{\Delta}$ and $\mathbf{S\Delta}$ are strongly equivalent (Corollary 1), meaning that $\mathbf{\Delta}$ is upper- and lower-bounded by $\mathbf{S\Delta}$.

Then, we derive the following statistical properties of $\mathbf{S\Delta}$: we prove that the 'sample complexity' of $\mathbf{S\Delta}$ is proportional to the sample complexity of $\mathbf{\Delta}$ for one-dimensional measures, and does not depend on the dimension $d$ (Theorems 4, 5). This property explains why *any* $\mathbf{S\Delta}$ motivated by the second purpose offers statistical benefits when the original divergence suffers from the curse of dimensionality. However, this comes with a caveat: we show that, if one approximates the expectation over the random projections that appears in $\mathbf{S\Delta}$ with a Monte Carlo average, which is the most common practice, then an additional variance term appears in the sample complexity and can limit the performance of $\mathbf{S\Delta}$ in high dimensions (Theorem 6). Our results agree with the recent empirical observations reported in [12, 15] and provide a better understanding for them.

We illustrate all our theoretical findings on various examples, which demonstrate their applicability. In particular, our general topological analysis allows us to establish a novel result for the Sliced-Cramér distance. We also derive a sample complexity result for SW which has never been shown before, under different assumptions on the measures to be compared. We then consider Sinkhorn divergences [22], whose sample complexity is known to have an exponential dependence on the dimension $d$ and regularization parameter $\varepsilon$ [23], and introduce its sliced version, referred to as the Sliced-Sinkhorn divergence. We prove that this new divergence has several merits: we derive its sample complexity by combining our general results with recent work [23, 24], and obtain rates that do not depend on $d$ nor on $\varepsilon$. We also show that this divergence improves the worst-case computational complexity bounds of Sinkhorn divergences in $\mathbb{R}^d$. Finally, we support our theory with numerical experiments on synthetic and real data.

## 2 Preliminaries and Technical Background

**Notations.** For $d \in \mathbb{N}^*$, let $\mathsf{X}$ be a closed and measurable subset of $\mathbb{R}^d$ and $\mathcal{B}(\mathsf{X})$ its Borel $\sigma$-algebra for the induced topology. $\mathcal{P}(\mathsf{X})$ stands for the set of probability measures on $(\mathsf{X}, \mathcal{B}(\mathsf{X}))$, and $\mathcal{P}_p(\mathsf{X}) = \left\{ \mu \in \mathcal{P}(\mathsf{X}) : \int_{\mathsf{X}} \|x\|^p \, \mathrm{d}\mu(x) < +\infty \right\}$ is the set of probability measures on $(\mathsf{X}, \mathcal{B}(\mathsf{X}))$ with finite moment of order $p$. Define for any $n \geq 1$, $\hat{\mu}_n$ the empirical distribution computed over a sequence of independent and identically distributed (i.i.d.) random variables $\{X_k\}_{k=1}^n$ sampled from $\mu$, by $\hat{\mu}_n = (1/n) \sum_{k=1}^n \delta_{X_k}$, with $\delta_x$ the Dirac measure at $x$. $\mathbb{M}(\mathsf{X})$ is the set of real-valued measurable functions on $\mathsf{X}$, and $\mathbb{M}_b(\mathsf{X})$ is the set of bounded functions of $\mathbb{M}(\mathsf{X})$. $\mathbb{S}^{d-1} = \left\{ \theta \in \mathbb{R}^d : \|\theta\| = 1 \right\}$ denotes the unit sphere in $\mathbb{R}^d$, and $\mathrm{B}_d(\mathbf{0}, R) = \left\{ x \in \mathbb{R}^d : \|x\| < R \right\}$ is the open ball in $\mathbb{R}^d$ of radius $R > 0$ centered around $\mathbf{0} \in \mathbb{R}^d$.

**Integral Probability Metrics.** For any measurable space $\mathsf{Y}$, let $\mathsf{F} \subset \mathbb{M}(\mathsf{Y})$ and $\mathcal{P}_{\mathsf{F}}(\mathsf{Y}) = \{ \mu \in \mathcal{P}(\mathsf{Y}) : \forall f \in \mathsf{F}, \int_{\mathsf{Y}} |f(y)| \, \mathrm{d}\mu(y) < +\infty \}$. The Integral Probability Metric (IPM, [21]) associated

with F and denoted by $\boldsymbol{\gamma}_{\mathsf{F}}$, is defined for any $\mu, \nu \in \mathcal{P}_{\mathsf{F}}(\mathsf{Y})$ as

$$\boldsymbol{\gamma}_{\mathsf{F}}(\mu, \nu) = \sup_{f \in \mathsf{F}} \left| \int_{\mathsf{Y}} f(y) \mathrm{d}(\mu - \nu)(y) \right| . \tag{1}$$

If $\mu$ or $\nu$ does not belong to $\mathcal{P}_{\mathsf{F}}(\mathsf{Y})$, we set $\boldsymbol{\gamma}_{\mathsf{F}}(\mu, \nu) = +\infty$. IPMs are pseudo-metrics [25]: they are non-negative, symmetric, satisfy the triangle inequality and for any $\mu \in \mathcal{P}_{\mathsf{F}}(\mathsf{Y})$, $\boldsymbol{\gamma}_{\mathsf{F}}(\mu, \mu) = 0$. We recall well-known instances of IPMs below.

(1) *Wasserstein distance of order 1.* By the Monge Kantorovich duality theorem [26, Theorem 5.10], when $\mathsf{F} = \{f : \mathsf{Y} \to \mathbb{R} : \|f\|_{\mathrm{Lip}} \leq 1\}$, where $\|f\|_{\mathrm{Lip}} = \sup_{x, y \in \mathsf{Y}, x \neq y}\{|f(x) - f(y)| / \|x - y\|\}$, $\boldsymbol{\gamma}_{\mathsf{F}}$ is the Wasserstein distance of order 1, denoted by $\mathbf{W}_1$.

(2) *Maximum mean discrepancy.* Let $\mathsf{H}$ be a reproducing kernel Hilbert space (RKHS) for real-valued functions on $\mathsf{Y}$, and $\mathsf{F}$ be the unit ball in $\mathsf{H}$. Then, $\boldsymbol{\gamma}_{\mathsf{F}}$ defines the MMD in RKHS [27, Section 2].

(3) *Cramér distance.* By [28, Lemma 1], the Cramér distance [9, Eq.(10)] can be written as an IPM.

In some of our results presented in Section 3, we will assume that the supremum in (1) is attained. This property is for example verified for $\mathbf{W}_1$ and MMD, by [26] and [27] respectively.

**Wasserstein distance, Sinkhorn divergences.** Arising from the optimal transportation (OT) theory, the Wasserstein distance of order $p \in [1, \infty)$ for any $\mu, \nu \in \mathcal{P}_p(\mathbb{R}^d)$ is defined as [26, Definition 6.1]

$$\mathbf{W}_p^p(\mu, \nu) = \inf_{\gamma \in \Gamma(\mu, \nu)} \int_{\mathbb{R}^d \times \mathbb{R}^d} \|x - y\|^p \, \mathrm{d}\gamma(x, y) , \tag{2}$$

where $\Gamma(\mu, \nu)$ represents the set of probability measures $\gamma$ on $\left( \mathbb{R}^d \times \mathbb{R}^d, \mathcal{B}(\mathbb{R}^d \otimes \mathbb{R}^d) \right)$ such that for any $\mathsf{A} \in \mathcal{B}(\mathbb{R}^d)$, $\gamma(\mathsf{A} \times \mathbb{R}^d) = \mu(\mathsf{A})$ and $\gamma(\mathbb{R}^d \times \mathsf{A}) = \nu(\mathsf{A})$. Note that, by strong duality [26, Theorem 5.10], $\mathbf{W}_1$ can be characterized by (2) or as an IPM (1).

When $\mu$ and $\nu$ are discrete distributions, computing $\mathbf{W}_p(\mu, \nu)$ amounts to solving a linear program, so its computational complexity becomes excessive in large-scale applications. By adding an entropic penalization term to (2), one can obtain an approximate solution using a simple numerical scheme with significantly lower computational requirements [29]. This yields a regularized Wasserstein cost: for any $\mu, \nu \in \mathcal{P}_p(\mathbb{R}^d)$ and $\varepsilon \geq 0$,

$$\mathbf{W}_{p,\varepsilon}(\mu, \nu) = \inf_{\gamma \in \Gamma(\mu, \nu)} \left\{ \int_{\mathbb{R}^d \times \mathbb{R}^d} \|x - y\|^p \, \mathrm{d}\gamma(x, y) + \varepsilon \mathbf{H}(\gamma \mid \mu \otimes \nu) \right\} , \tag{3}$$

where $\mathbf{H}(\gamma \mid \mu \otimes \nu)$ is the relative entropy of the transport plan $\gamma$ with respect to $\mu \otimes \nu$: if $\gamma$ is absolutely continuous with respect to $\mu \otimes \nu$, $\mathbf{H}(\gamma \mid \mu \otimes \nu) = \int_{\mathbb{R}^d \times \mathbb{R}^d} \log[(\mathrm{d}\gamma/\mathrm{d}\mu \otimes \nu)(x, y)] \mathrm{d}\gamma(x, y)$, otherwise, $\mathbf{H}(\gamma \mid \mu \otimes \nu) = +\infty$. Building on the regularized Wasserstein cost, [22] defined Sinkhorn divergences for $\mu, \nu \in \mathcal{P}_p(\mathbb{R}^d)$ and $\varepsilon \geq 0$ as $\overline{\mathbf{W}}_{p,\varepsilon}(\mu, \nu) = \mathbf{W}_{p,\varepsilon}(\mu, \nu) - \{\mathbf{W}_{p,\varepsilon}(\mu, \mu) + \mathbf{W}_{p,\varepsilon}(\nu, \nu)\}/2$. These satisfy for any $\mu \in \mathcal{P}_p(\mathbb{R}^d)$, $\overline{\mathbf{W}}_{p,\varepsilon}(\mu, \mu) = 0$ (contrary to $\mathbf{W}_{p,\varepsilon}$), and interpolate between OT (when $\varepsilon \to 0$) and MMD ($\varepsilon \to \infty$).

**Sliced-Wasserstein (SW) distance.** When dealing with one-dimensional distributions, (2) admits a closed-form solution, which can be efficiently computed. This gave rise to another popular tool called SW, which has been successfully used for generative modeling applications [20, 30, 31, 32]. The main idea is to consider one-dimensional *linear projections* of two high-dimensional measures, then compute the expected $\mathbf{W}_p$ between these representations. Formally, the Sliced-Wasserstein distance of order $p \in [1, \infty)$, is defined for any $\mu, \nu \in \mathcal{P}_p(\mathbb{R}^d)$ as

$$\mathbf{SW}_p^p(\mu, \nu) = \int_{\mathbb{S}^{d-1}} \mathbf{W}_p^p(\theta_\sharp^\star \mu, \theta_\sharp^\star \nu) \mathrm{d}\boldsymbol{\sigma}(\theta) ,$$

where $\boldsymbol{\sigma}$ is the uniform distribution on $\mathbb{S}^{d-1}$, and for any $\theta \in \mathbb{S}^{d-1}$, $\theta^\star : \mathbb{R}^d \to \mathbb{R}$ denotes the linear form given by $x \mapsto \langle \theta, x \rangle$ with $\langle \cdot, \cdot \rangle$ the Euclidean inner-product. For any measurable function $f : \mathbb{R}^d \to \mathbb{R}$ and $\zeta \in \mathcal{P}(\mathbb{R}^d)$, $f_\sharp \zeta$ is the push-forward measure of $\zeta$ by $f$, *i.e.* for any $\mathsf{A} \in \mathcal{B}(\mathbb{R})$, $f_\sharp \zeta(\mathsf{A}) = \zeta(f^{-1}(\mathsf{A}))$, with $f^{-1}(\mathsf{A}) = \{x \in \mathbb{R}^d : f(x) \in \mathsf{A}\}$,

## 3 Sliced Probability Divergences

In this section, we define the family of Sliced Probability Divergences (SPDs), then we present our theoretical contributions regarding their topological and statistical properties. We provide all the proofs in the supplementary document.

Consider a *'base divergence'* $\boldsymbol{\Delta} : \mathcal{P}(\mathbb{R}) \times \mathcal{P}(\mathbb{R}) \to \mathbb{R}_+ \cup \{\infty\}$ which measures the dissimilarity between two probability measures on $\mathbb{R}$. We define the Sliced Probability Divergence of order $p \in [1, \infty)$ associated to $\boldsymbol{\Delta}$, denoted by $\mathbf{S}\boldsymbol{\Delta}_p$, for $\mu, \nu \in \mathcal{P}(\mathbb{R}^d)$ as

$$\mathbf{S}\boldsymbol{\Delta}_p^p(\mu, \nu) = \int_{\mathbb{S}^{d-1}} \boldsymbol{\Delta}^p(\theta_\sharp^\star \mu, \theta_\sharp^\star \nu) \mathrm{d}\boldsymbol{\sigma}(\theta) . \tag{4}$$

We assume that $\theta \mapsto \boldsymbol{\Delta}^p(\theta_\sharp^\star \mu, \theta_\sharp^\star \nu)$ is measurable so that (4) is well-defined. This can easily be checked if $(\mu', \nu') \mapsto \boldsymbol{\Delta}(\mu', \nu')$ is continuous for the weak topology on $\mathcal{P}(\mathbb{R})$, since this implies $\theta \mapsto \boldsymbol{\Delta}^p(\theta_\sharp^\star \mu, \theta_\sharp^\star \nu)$ is continuous.

In practice, since the integration over $\mathbb{S}^{d-1}$ in (4) does not admit an analytical form in general, it is approximated with a simple Monte Carlo scheme (e.g., [8, 15, 16, 9]). The Monte Carlo estimate of $\mathbf{S}\boldsymbol{\Delta}_p$ obtained with $L$ random projection directions is defined as

$$\widehat{\mathbf{S}\boldsymbol{\Delta}}_{p,L}^p(\mu, \nu) = (1/L) \sum_{l=1}^L \boldsymbol{\Delta}^p(\theta_{l\sharp}^\star \mu, \theta_{l\sharp}^\star \nu) , \tag{5}$$

with $\{\theta_l\}_{l=1}^L$ i.i.d. from $\boldsymbol{\sigma}$ and $\theta_l^\star(x) = \langle \theta_l, x \rangle$. Since each term of the sum in (5) can be computed independently from each other, the approximation of SPDs can be carried out in parallel, which constitutes a nice practical feature. Recent work [13, 12] has shown that sampling many projection directions uniformly on the sphere might not be the best strategy, in the sense that some directions can be more helpful than others to discriminate the two distributions at hand. However, the Monte Carlo estimate based on uniform sampling (5) is the most common method used in practice to approximate sliced divergences, hence we focus on this approximation throughout the rest of the paper.

**Topological properties.** We provide several results to describe the topology induced by SPDs, given the properties of base divergences. We first relate the metric properties of $\boldsymbol{\Delta}$ and $\mathbf{S}\boldsymbol{\Delta}_p$, $p \in [1, \infty)$.

**Proposition 1.** *(i) If $\boldsymbol{\Delta}$ is non-negative (or symmetric), then $\mathbf{S}\boldsymbol{\Delta}_p$ is non-negative (symmetric resp.).*
*(ii) If $\boldsymbol{\Delta}$ satisfies for $\mu', \nu' \in \mathcal{P}(\mathbb{R})$, $\boldsymbol{\Delta}(\mu', \nu') = 0$ if and only if $\mu' = \nu'$, then for $\mu, \nu \in \mathcal{P}(\mathbb{R}^d)$, $\mathbf{S}\boldsymbol{\Delta}_p(\mu, \nu) = 0$ if and only if $\mu = \nu$.*
*(iii) If $\boldsymbol{\Delta}$ is a metric, then $\mathbf{S}\boldsymbol{\Delta}_p$ is a metric.*

Next, we extend the result in [18, Theorem 1], which showed that the convergence in SW implies the weak convergence of probability measures: we prove that this property holds for the general class of SPDs, but also that the converse implication is true, provided that $\boldsymbol{\Delta}$ is weakly continuous. Before presenting this result in Theorem 1, we recall the definitions of convergence under a probability divergence and weak convergence of probability measures.

**Definition 1.** *Let $d \in \mathbb{N}^*$ and $\mathbf{D} : \mathcal{P}(\mathbb{R}^d) \times \mathcal{P}(\mathbb{R}^d) \to \mathbb{R}_+ \cup \{\infty\}$ be a probability divergence. Let $(\mu_k)_{k \in \mathbb{N}}$ be a sequence in $\mathcal{P}(\mathbb{R}^d)$ and $\mu \in \mathcal{P}(\mathbb{R}^d)$. We introduce two types of convergence below.*

   **C1.** *"$(\mu_k)_{k \in \mathbb{N}}$ converges to $\mu$ under $\mathbf{D}$", i.e. $\lim_{k \to +\infty} \mathbf{D}(\mu_k, \mu) = 0$.*

   **C2.** *"$(\mu_k)_{k \in \mathbb{N}}$ converges weakly to $\mu$", i.e. for any continuous and bounded function $f : \mathbb{R}^d \to \mathbb{R}$, $\lim_{k \to +\infty} \int f \mathrm{d}\mu_k = \int f \mathrm{d}\mu$.*

*Hence, the statement "the convergence under $\mathbf{D}$ implies the convergence in $\mathcal{P}(\mathbb{R}^d)$" is equivalent to, C1 implies C2 for any $(\mu_k)_{k \in \mathbb{N}}$ and $\mu$ in $\mathcal{P}(\mathbb{R}^d)$. Conversely, "the weak convergence in $\mathcal{P}(\mathbb{R}^d)$ implies the convergence in $\mathbf{D}$" means that C2 implies C1 for any $(\mu_k)_{k \in \mathbb{N}}$ and $\mu$ in $\mathcal{P}(\mathbb{R}^d)$.*

**Theorem 1.** *Let $p \in [1, \infty)$ and $\boldsymbol{\Delta}$ be a non-negative base divergence.*
*(i) If the convergence under $\boldsymbol{\Delta}$ implies the weak convergence in $\mathcal{P}(\mathbb{R})$, then the convergence under $\mathbf{S}\boldsymbol{\Delta}_p$ implies the weak convergence in $\mathcal{P}(\mathbb{R}^d)$.*
*(ii) If $\boldsymbol{\Delta}$ is bounded and the weak convergence in $\mathcal{P}(\mathbb{R})$ implies the convergence under $\boldsymbol{\Delta}$, then the weak convergence in $\mathcal{P}(\mathbb{R}^d)$ implies the convergence under $\mathbf{S}\boldsymbol{\Delta}_p$.*

We now focus on IPMs and formally define Sliced-IPMs, before providing finer topological results.

**Definition 2.** *Let $\widetilde{\mathsf{F}} \subset \mathbb{M}_b(\mathbb{R})$, $p \in [1, \infty)$. The Sliced Integral Probability Metric of order $p$ associated with $\widetilde{\mathsf{F}}$, denoted by $\mathbf{S}\boldsymbol{\gamma}_{\widetilde{\mathsf{F}},p}$, is, for $\mu, \nu \in \mathcal{P}(\mathbb{R}^d)$, $(\mathbf{S}\boldsymbol{\gamma}_{\widetilde{\mathsf{F}},p})^p(\mu, \nu) = \int_{\mathbb{S}^{d-1}} \boldsymbol{\gamma}_{\widetilde{\mathsf{F}}}^p(\theta_\sharp^\star \mu, \theta_\sharp^\star \nu) \mathrm{d}\boldsymbol{\sigma}(\theta)$.*

Since $\gamma_{\widetilde{\mathsf{F}}}$ is a pseudo-metric, $\mathbf{S}\boldsymbol{\gamma}_{\widetilde{\mathsf{F}},p}$ is a pseudo-metric as well by Proposition 1. We now identify some regularity conditions on the function classes $\mathsf{F}$ and $\widetilde{\mathsf{F}}$ such that we are able to show that Sliced-IPMs can be bounded above and below by IPMs.

**Theorem 2.** *Let* $\widetilde{\mathsf{F}} \subset \mathbb{M}_b(\mathbb{R})$, $\mathsf{F} \subset \mathbb{M}_b(\mathbb{R}^d)$, $\left\{ f : \mathbb{R}^d \to \mathbb{R} \; : \; f = \tilde{f} \circ \theta^\star, \text{ with } \tilde{f} \in \widetilde{\mathsf{F}}, \theta \in \mathbb{S}^{d-1} \right\} \subset$ $\mathsf{F}$. *Then, for any* $p \in [1, \infty)$ *and* $\mu, \nu \in \mathcal{P}(\mathbb{R}^d)$, $\mathbf{S}\boldsymbol{\gamma}_{\widetilde{\mathsf{F}},p}(\mu, \nu) \leq \boldsymbol{\gamma}_{\mathsf{F}}(\mu, \nu)$.

Theorem 2 states that $\mathbf{S}\boldsymbol{\gamma}_{\widetilde{\mathsf{F}},p}$ induces a weaker topology, which is computationally beneficial as argued in [3], but also indicates that $\mathbf{S}\boldsymbol{\gamma}_{\widetilde{\mathsf{F}},p}$ comes with less discriminative power, which can be restrictive for hypothesis testing applications [27]. We now derive a lower-bound on compact domains.

**Theorem 3.** *Let* $\mu, \nu \in \mathcal{P}(\mathbb{R}^d)$, *with support included in* $\mathrm{B}_d(\mathbf{0}, R)$. *Let* $\mathsf{G} \subset \mathbb{M}_b(\mathbb{R}^d)$ *and suppose that there exists* $\mathsf{L} \geq 0$ *such that for any* $g \in \mathsf{G}$, $g$ *is* $\mathsf{L}$-*Lipschitz continuous. Consider a class of functions* $\widetilde{\mathsf{G}}$ *satisfying* $\widetilde{\mathsf{G}} \supset \{\tilde{g} : \mathbb{R} \to \mathbb{R} \; : \; \text{there exist } x \in \mathbb{R}^d, \theta \in \mathbb{S}^{d-1}, g \in \mathsf{G} \text{ such that } \tilde{g}(t) = g(x - \theta t) \text{ for any } t \in \mathbb{R}\}$. *Furthermore, suppose that* $\mathbf{S}\boldsymbol{\gamma}_{\widetilde{\mathsf{G}},p}$ *is bounded. Then, for any* $p \in [1, +\infty)$, *there exists* $C_p > 0$ *such that* $\boldsymbol{\gamma}_{\mathsf{G}}(\mu, \nu) \leq C_p \, \mathbf{S}\boldsymbol{\gamma}_{\widetilde{\mathsf{G}},p}(\mu, \nu)^{1/(d+1)}$.

One can show that the exponent $1/(d+1)$ is intrinsic to slicing and hence cannot be avoided. By combining the two theorems, we finally establish a strong equivalence result below, which implies that the convergence of probability measures in $\mathbf{S}\boldsymbol{\gamma}_{\widetilde{\mathsf{G}},p}$ is equivalent to the convergence in $\boldsymbol{\gamma}_{\mathsf{G}}$.

**Corollary 1.** *Let* $\mu, \nu \in \mathcal{P}(\mathbb{R}^d)$, *with support included in* $\mathrm{B}_d(\mathbf{0}, R)$, *and let* $\mathsf{G} \subset \mathbb{M}_b(\mathbb{R}^d)$. *Assume that the conditions of Theorem 3 are satisfied. Then, for any* $p \in [1, +\infty)$, *there exists* $C_p \geq 0$ *independent of* $\mu, \nu$ *such that* $\mathbf{S}\boldsymbol{\gamma}_{\widetilde{\mathsf{G}},p}(\mu, \nu) \leq \boldsymbol{\gamma}_{\mathsf{G}}(\mu, \nu) \leq C_p \, \mathbf{S}\boldsymbol{\gamma}_{\widetilde{\mathsf{G}},p}(\mu, \nu)^{1/(d+1)}$.

Our analysis on IPMs builds on [17, Chapter 5.1], which contains analogous results for the Sliced-Wasserstein distance only. The novelty of Theorems 2 and 3 is the identification of the relationships between the function classes $\widetilde{\mathsf{F}}, \mathsf{F}$ and $\widetilde{\mathsf{G}}, \mathsf{G}$, which might provide a useful guideline for practitioners interested in slicing any IPM, and cannot be directly obtained from [17]. We further illustrate these relations in the supplementary document for classical instances of IPMs.

**Statistical properties.** In most practical applications, we have at hand finite sets of samples drawn from unknown underlying distributions. An important question is then the bound of the error made when approximating a divergence with finitely many samples: given $\mathbf{S}\boldsymbol{\Delta}_p$ and any $\mu, \nu \in \mathcal{P}(\mathbb{R}^d)$, our goal is to quantify the *sample complexity* of $\mathbf{S}\boldsymbol{\Delta}_p$, *i.e.* the convergence rate of $\mathbf{S}\boldsymbol{\Delta}_p(\hat{\mu}_n, \hat{\nu}_n)$ to $\mathbf{S}\boldsymbol{\Delta}_p(\mu, \nu)$ according to $n$. We show that the sample complexity of any SPD is proportional to the sample complexity of the base divergence, and more importantly, does not depend on $d$.

**Theorem 4.** *Let* $p \in [1, \infty)$. *Suppose that* $\boldsymbol{\Delta}^p$ *admits the following sample complexity: for any* $\mu', \nu' \in \mathcal{P}(\mathbb{R})$ *with respective empirical measures* $\hat{\mu}'_n, \hat{\nu}'_n$, $\mathbb{E}\left|\boldsymbol{\Delta}^p(\mu', \nu') - \boldsymbol{\Delta}^p(\hat{\mu}'_n, \hat{\nu}'_n)\right| \leq \beta(p, n)$. *Then, for any* $\mu, \nu \in \mathcal{P}(\mathbb{R}^d)$ *with respective empirical measures* $\hat{\mu}_n, \hat{\nu}_n$, *the sample complexity of* $\mathbf{S}\boldsymbol{\Delta}_p$ *is given by* $\mathbb{E}\left|\mathbf{S}\boldsymbol{\Delta}_p^p(\mu, \nu) - \mathbf{S}\boldsymbol{\Delta}_p^p(\hat{\mu}_n, \hat{\nu}_n)\right| \leq \beta(p, n)$.

If $\boldsymbol{\Delta}$ is a bounded pseudo-metric and we have a direct control over the convergence rate of empirical measures in $\boldsymbol{\Delta}$, we can further derive the following result.

**Theorem 5.** *Let* $p \in [1, \infty)$. *Assume that for any* $\mu' \in \mathcal{P}(\mathbb{R})$ *with empirical measure* $\hat{\mu}'_n$, $\mathbb{E}\left|\boldsymbol{\Delta}^p(\hat{\mu}'_n, \mu')\right| \leq \alpha(p, n)$. *Then, for any* $\mu \in \mathcal{P}(\mathbb{R}^d)$ *with empirical measure* $\hat{\mu}_n$, *we have* $\mathbb{E}\left|\mathbf{S}\boldsymbol{\Delta}_p^p(\hat{\mu}_n, \mu)\right| \leq \alpha(p, n)$. *Besides, if* $\boldsymbol{\Delta}$ *is non-negative, symmetric, and satisfies the triangle inequality, then* $\mathbb{E}\left|\mathbf{S}\boldsymbol{\Delta}_p(\mu, \nu) - \mathbf{S}\boldsymbol{\Delta}_p(\hat{\mu}_n, \hat{\nu}_n)\right| \leq 2\,\alpha(p, n)^{1/p}$.

Our results show that slicing leads to a dimension-free convergence rate while carrying out useful topological properties of the base divergence (e.g., metric axioms, weak convergence). If the focus is on sustaining such topological properties, then the improvement in the convergence rate is meaningful. On the other hand, slicing also results in less discriminant divergences, as we mentioned for IPMs (Theorem 2), and in such a case, the improvement in the rate might be less significant. More analysis is required to understand the potential reduction in the discriminative power, and we leave it out of scope of this study.

In practice, SPDs also induce an approximation error due to the Monte Carlo estimate (5). We use the term *projection complexity* to refer to the convergence rate of $\widehat{\mathbf{S}\boldsymbol{\Delta}}_{p,L}$ to $\mathbf{S}\boldsymbol{\Delta}_p$ as a function of the number of projections $L$. Hence, the *overall complexity* $\left|\widehat{\mathbf{S}\boldsymbol{\Delta}}_{p,L}(\hat{\mu}_n, \hat{\nu}_n) - \mathbf{S}\boldsymbol{\Delta}_p(\mu, \nu)\right|$ is bounded by the sum of the sample and the projection complexities.

**Theorem 6.** *Let $p \in [1, \infty)$ and $\mu, \nu \in \mathcal{P}(\mathbb{R}^d)$. Then, the error made with the Monte Carlo estimation of $\mathbf{S\Delta}_p$ can be bounded as follows*

$$\left\{ \mathbb{E} \left| \widehat{\mathbf{S\Delta}}_{p,L}^p(\mu, \nu) - \mathbf{S\Delta}_p^p(\mu, \nu) \right| \right\}^2 \leq L^{-1} \int_{\mathbb{S}^{d-1}} \left\{ \mathbf{\Delta}^p(\theta_\sharp^\star \mu, \theta_\sharp^\star \nu) - \mathbf{S\Delta}_p^p(\mu, \nu) \right\}^2 \mathrm{d}\boldsymbol{\sigma}(\theta) .$$

By definition of $\mathbf{S\Delta}_p^p(\mu, \nu)$, Theorem 6 illustrates that the quality of the Monte Carlo estimates is impacted by the number of projections as well as the variance of the evaluations of the base divergence. This behavior has previously been empirically observed in different scenarios [12, 15, 13], and paved the way for the 'max-sliced' distances. We additionally provide in the supp. document, finite-sample guarantees on the quality of the Monte Carlo estimates, using Theorems 4 and 6.

## 4    Applications

In this section, to further illustrate the significance of our general topological and statistical results, we apply these to specific sliced divergences and present the interesting properties that we obtained. In particular, we will introduce a novel divergence based on Sinkhorn divergences, and provide theoretical results that emphasize its statistical and computational advantages.

First, Theorem 1 can be applied to various base divergences (e.g., see those listed in [33, Theorem 6]) and foster interesting applications. In particular, we focus on the Sliced-Cramér distance (SC, [11, 9]) and establish theoretical guarantees which, to the best of our knowledge, have not been proved before: we show that convergence under SC implies weak convergence in $\mathcal{P}(\mathbb{R}^q)$, and the converse is true for measures supported on a compact space. Our general result also applies to the broader class of Sliced-IPMs, assuming a density property for the space of functions associated with the base IPM. We provide the formal statements and proofs of these results in the supplementary document.

Then, we derive the sample complexity of $\mathbf{SW}_p$ under different moment conditions. While previous works have illustrated the statistical benefits of SW, our next corollary establishes a novel result: [12] derived the sample complexity for Gaussian distributions only, [18] studied the estimators obtained by minimizing SW, and [34] provided confidence intervals which partially cover our result.

**Corollary 2.** *Let $p \in [1, \infty)$, $q > p$, and $\mu, \nu \in \mathcal{P}_q(\mathbb{R}^d)$ with corresponding empirical measures $\hat{\mu}_n, \hat{\nu}_n$. We use the notation $M_q^{1/q}(\mu, \nu) = M_q^{1/q}(\mu) + M_q^{1/q}(\nu)$, where $M_q(\zeta)$ refers to the moment of order $q$ of $\zeta \in \mathcal{P}_q(\mathbb{R}^d)$. Then, there exists a constant $C_{p,q}$ depending on $p, q$ such that*

$$\mathbb{E}\left| \mathbf{SW}_p(\hat{\mu}_n, \hat{\nu}_n) - \mathbf{SW}_p(\mu, \nu) \right| \leq C_{p,q}^{1/p} M_q^{1/q}(\mu, \nu) \begin{cases} n^{-1/(2p)} & \text{if } q > 2p, \\ n^{-1/(2p)} \log(n)^{1/p} & \text{if } q = 2p, \\ n^{-(q-p)/(pq)} & \text{if } q \in (p, 2p), \end{cases}$$

We now introduce a new family of probability divergences obtained by slicing the regularized OT cost and Sinkhorn divergences, and called Sliced-Sinkhorn divergences (SSD): for $p \in [1, \infty)$, $\varepsilon \geq 0$ and $\mu, \nu \in \mathcal{P}_p(\mathbb{R}^d)$,

$$\mathbf{SW}_{p,\varepsilon}(\mu, \nu) = \int_{\mathbb{S}^{d-1}} \mathbf{W}_{p,\varepsilon}(\theta_\sharp^\star \mu, \theta_\sharp^\star \nu) \, \mathrm{d}\boldsymbol{\sigma}(\theta), \quad \overline{\mathbf{SW}}_{p,\varepsilon}(\mu, \nu) = \int_{\mathbb{S}^{d-1}} \overline{\mathbf{W}}_{p,\varepsilon}(\theta_\sharp^\star \mu, \theta_\sharp^\star \nu) \, \mathrm{d}\boldsymbol{\sigma}(\theta) \quad (6)$$

We show that these divergences enjoy interesting statistical and computational properties. For clarity purposes, our results are only presented for $\mathbf{SW}_{p,\varepsilon}$, but also apply for $\overline{\mathbf{SW}}_{p,\varepsilon}$. Since $\mathbf{W}_{p,\varepsilon}$ is not an IPM, we first derive a topological property analogous to Theorem 2.

**Theorem 7.** *Let $p \in [1, \infty)$ and $\varepsilon \geq 0$. For any $\mu, \nu \in \mathcal{P}_p(\mathbb{R}^d)$, $\mathbf{SW}_{p,\varepsilon}(\mu, \nu) \leq \mathbf{W}_{p,\varepsilon}(\mu, \nu)$.*

Next, we show that on compact domains, while the sample complexity of regularized OT exponentially worsens as $\varepsilon$ decreases [23, Theorem 3], the sample complexity of SSD does not depend on $\varepsilon$.

**Theorem 8.** *Let $\mathsf{X}$ be a compact subset of $\mathbb{R}^d$, $p \in [1, \infty)$ and $\mu, \nu \in \mathcal{P}(\mathsf{X})$, with respective empirical instanciations $\hat{\mu}_n, \hat{\nu}_n$. Then, there exists a constant $C(\mu, \nu)$ that depends on the moments of $\mu$ and $\nu$, such that $\mathbb{E}\left| \mathbf{SW}_{p,\varepsilon}(\hat{\mu}_n, \hat{\nu}_n) - \mathbf{SW}_{p,\varepsilon}(\mu, \nu) \right| \leq \mathrm{diam}(\mathsf{X}) C(\mu, \nu) n^{-1/2}$.*

In practice, we approximate SSD by using (5). The estimator corresponds to randomly picking a finite set of directions and solving, for each direction, a regularized OT problem in $\mathbb{R}$. To obtain solutions associated to the regularized Wasserstein cost (3), a method which is now standard is the

Sinkhorn's algorithm ([35]; more details in the supp. document, [36, Section 4.2]). In particular, if we use the squared Euclidean ground cost and consider the empirical measures $\hat{\mu}_n, \hat{\nu}_n$ on $\mathbb{R}^d$ associated to the observations $(x_i)_{i=1}^n, (y_j)_{j=1}^n$ respectively, computing $\mathbf{W}_{p,\varepsilon}(\hat{\mu}_n, \hat{\nu}_n)$ has a worst-case convergence rate that depends on $C(\hat{\mu}_n, \hat{\nu}_n) = \max_{i,j \in \{1,...,n\}} \|x_i - y_j\|^2/\varepsilon$ (see also [37] for a sublinear rate with a better constant, still depending on this quantity). The rate for $\mathbf{W}_{p,\varepsilon}(\theta_\sharp^\star \hat{\mu}_n, \theta_\sharp^\star \hat{\nu}_n)$, with $\theta \in \mathbb{S}^{d-1}$, then depends on $C(\theta_\sharp^\star \hat{\mu}_n, \theta_\sharp^\star \hat{\nu}_n) = \max_{i,j \in \{1,...,n\}} \|\langle\theta, x_i - y_j\rangle\|^2/\varepsilon$. We show that with high probability, $C(\theta_\sharp^\star \hat{\mu}_n, \theta_\sharp^\star \hat{\nu}_n)$ is smaller than $C(\hat{\mu}_n, \hat{\nu}_n)$ by a factor of $d$ at least, unless $n$ grows super-polynomially with $d$. Our result, combined with the parallel computation of (5), implies that slicing the regularized OT may lead to significant computational benefits.

**Proposition 2.** *Let $(x_i)_{i=1}^n$ be a set of vectors in $\mathbb{R}^d$ such that $\max_{i,j} \|x_i - x_j\|_2^2 \leq R^2$, and $\theta$ chosen uniformly at random on $\mathbb{S}^{d-1}$. Then for $\delta \in (0,1]$, it holds with probability $1 - \delta$, $\max_{i,j} |\langle\theta, x_i - x_j\rangle|^2 \leq \frac{2R^2}{d} \log(\sqrt{2\pi}n^2/\delta)$.*

Finally, we note that an advantage of the Sinkhorn divergence over the Wasserstein distance is that the former is always differentiable [22, Proposition 2] while the latter is not. This property, which is crucial in differential programming pipelines, suggests that SSD is potentially better-behaved than SW in tasks such as generative modeling. We leave its analysis to future work.

## 5    Experiments

We present the numerical experiments that we conducted to illustrate our theoretical findings, and we provide the code to reproduce them[2].

We first verify that IPMs and Sinkhorn divergences are bounded below by their sliced versions, as demonstrated in Theorems 2 and 7 respectively. Consider $n = 1000$ observations i.i.d. from $\mathcal{N}(\mathbf{0}, \sigma_\star^2 \mathbf{I}_d)$, with $\sigma_\star^2 = 4$. We generate $n$ i.i.d. samples from $\mathcal{N}(\mathbf{0}, \sigma^2 \mathbf{I}_d)$ for $\sigma^2$ varying between 0.1 and 9. We compute MMD between the empirical distributions of the observations and the generated datasets, as well as the Wasserstein distance of order 1 and normalized Sinkhorn divergence (6) with order 1 and $\varepsilon = 1$. We used a Gaussian kernel for MMD combined with the heuristic proposed in [27], which sets the kernel width to be the median distance over the aggregated data, and we approximated this discrepancy with the biased estimator in [27, Equation 5]. Then, we compute Sliced-Wasserstein, Sliced-Sinkhorn and Sliced-MMD. Each of these sliced divergences was approximated with a Monte Carlo estimate based on 50 randomly picked projections. Figure 1 reports the divergences against $\sigma^2$ for $d = 10$. Results are aver-

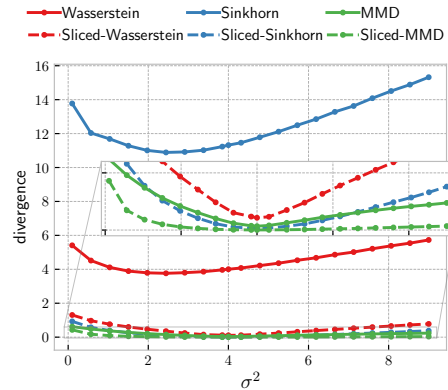

Figure 1: (Sliced-)Divergences between two sets of 1000 samples in $\mathbb{R}^{10}$ i.i.d. from $\mathcal{N}(\mathbf{0}, 4\mathbf{I})$ and $\mathcal{N}(\mathbf{0}, \sigma^2 \mathbf{I})$, for varying $\sigma^2$.

aged over 10 runs, and for clarity reasons, we do not plot the error bands (based on the 10th-90th percentiles) as these were very tight. The curves for Wasserstein, Sinkhorn and MMD are above their respective sliced version's ones, as predicted by our theoretical bounds. This figure also illustrates the statistical benefits induced by slicing: all sliced divergences attain their minimum at $\sigma_\star$, while Wasserstein and Sinkhorn fail at this. This observation is in line with [38], where the authors showed that both the minimum point and gradients of the Wasserstein distance have a bias, which can be prominent unless $n$ is large enough. MMD performs well in this task, and this can be explained by its dimension-free sample complexity. In that sense, Sliced-MMD acts more as a sanity-check of our theory, rather than a practical proposal.

The next experiments aim at illustrating our statistical properties We first analyze the convergence rate of the Monte Carlo estimates (Theorem 6) in a synthetical setting. We consider two sets of 500 samples i.i.d. from the $d$-dimensional Gaussian distribution $\mathcal{N}(\mathbf{0}, \mathbf{I}_d)$, and we approximate $\mathbf{SW}_2$ between the empirical distributions with a Monte Carlo scheme that uses a high number of projections $L_\star = 10\,000$. Then, we compute the Monte Carlo estimate $\widehat{\mathbf{SW}}_{2,L}$ obtained with $L < L_\star$ random

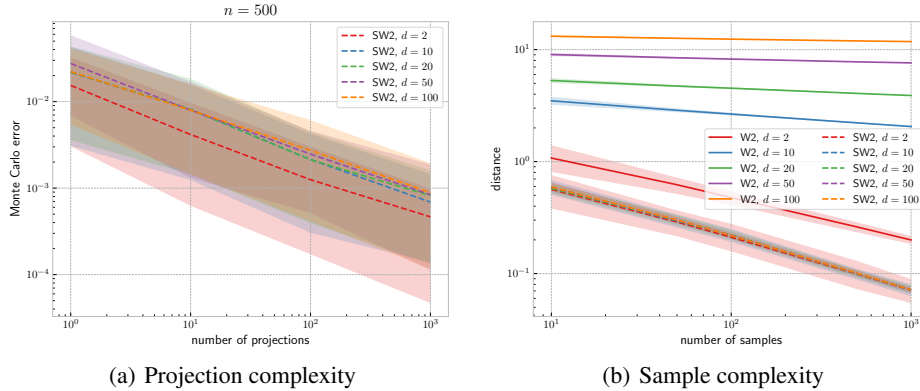

(a) Projection complexity          (b) Sample complexity

Figure 2: (Sliced-)Wasserstein distances of order 2 between two sets of $n$ samples generated from $\mathcal{N}(\mathbf{0}, \mathbf{I}_d)$ for different $d$, on log-log scale. Results are averaged over 100 runs, and the shaded areas correspond to the 10th-90th percentiles.

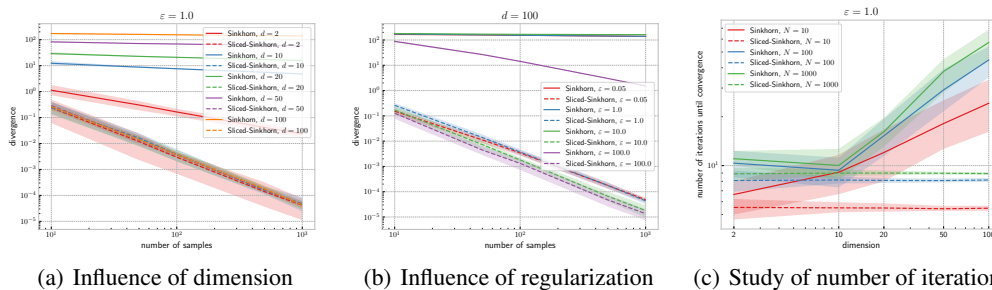

(a) Influence of dimension    (b) Influence of regularization    (c) Study of number of iterations

Figure 3: (Sliced-)Sinkhorn divergences between two sets of $n$ samples generated from $\mathcal{N}(\mathbf{0}, \mathbf{I}_d)$ for different values of $n$, dimension $d$, and regularization coefficient $\varepsilon$. Results are averaged over 100 runs, and the shaded areas correspond to the 10th-90th percentiles. All plots have a log-log scale.

projections. Figure 2(a) shows the absolute difference of $\widehat{\mathbf{SW}}_{2,L}$ and $\widehat{\mathbf{SW}}_{2,L_\star}$ against $L$, for different values of dimension $d$. We observe that the Monte Carlo error indeed shrinks to zero when we increase the number of projections, with a convergence rate of order $L^{-1/2}$.

Then, we illustrate the sample complexity of Sliced-Wasserstein and Sliced-Sinkhorn (Corollary 2 and Theorem 8, respectively). We consider two sets of $n$ samples i.i.d. from $\mathcal{N}(\mathbf{0}, \mathbf{I}_d)$, and we compute $\mathbf{W}_2$ and $\overline{\mathbf{W}}_{2,\varepsilon}$ and their sliced versions approximated with 100 random projections. We analyze the convergence rate for different $n$ and dimensions $d$. We also study the influence of the regularization parameter $\varepsilon$ for Sinkhorn divergences. Figure 2(b) reports the Wasserstein and Sliced-Wasserstein distances vs. $n$, for $d$ between 2 and 100. We observe that, as opposed to $\mathbf{W}_2$, the convergence rate of $\mathbf{SW}_2$ does not depend on the dimension, therefore $\mathbf{SW}_2$ converges faster than $\mathbf{W}_2$ when the dimension increases. Figures 3(a) and 3(b) show Sinkhorn and Sliced-Sinkhorn divergences vs. $n$, and respectively study the influence of $d$ and $\varepsilon$ on the convergence rate. As predicted by the theory, Sliced-Sinkhorn offers more 'robustness' than Sinkhorn: its convergence rate does not depend on the dimension nor on the regularization coefficient. To illustrate Proposition 2, we plot on Figure 3(c) the number of iterations when the convergence of Sinkhorn's algorithm is reached, as a function of $d$. For Sliced-Sinkhorn, this number is an average over the number of projections used in the approximation. Our experiment emphasizes the computational advantages of Sliced-Sinkhorn, since its number of iterations remains the same with the increasing dimension, while it grows exponentially for Sinkhorn.

Our last experiment operates on real data and is motivated by the two-sample testing problem [27], whose goal is to determine whether two sets of samples were generated from the same distribution or not. This is useful for various applications, including data integration, where we wish to understand that two datasets were drawn from the same distribution in order to merge them. In this context, we run the following experiment: for different values of $n$, we randomly select two subsets of $n$ samples from the same dataset, and we compute the Wasserstein and Sliced-Wasserstein distances (of order 2) between the empirical distributions, as well as the Sinkhorn and Sliced-Sinkhorn divergences

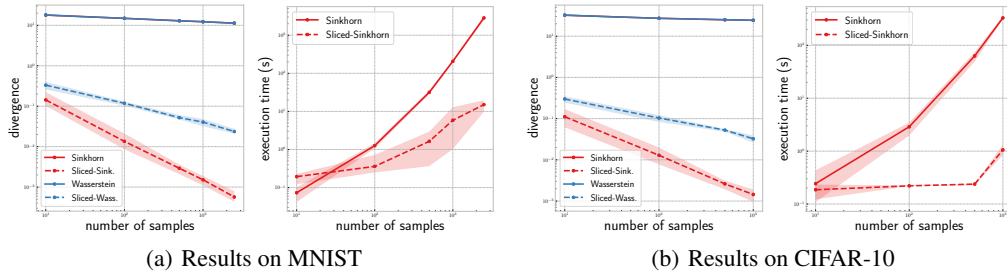

| (a) Results on MNIST | (b) Results on CIFAR-10 |

Figure 4: (Sliced-)Wasserstein and (Sliced-)Sinkhorn ($\varepsilon = 1$) between two random subsets of $n$ samples of real datasets, for different values of $n$. Results are averaged over 10 runs, and the shaded areas correspond to the 10th-90th percentiles. All plots have a log-log scale.

($\varepsilon = 1$). The sliced divergences are approximated with 10 random projections. We use the MNIST [39] and CIFAR-10 [40, Chapter 3] datasets, and we report the divergences against $n$, and the mean execution time for the computation of Sinkhorn and Sliced-Sinkhorn, on Figure 4. The sliced divergences perform the best, in the sense that they need less samples to converge to zero. Besides, Sliced-Sinkhorn is faster than Sinkhorn in terms of execution time (which was expected, given our discussion above Proposition 2), and the difference is even more visible for a high number of samples. For example, for $n = 2500$ on MNIST or $n = 1000$ on CIFAR-10, Sliced-Sinkhorn is almost 130 times faster than for Sinkhorn on average.

## 6 Conclusion

In this study, we considered sliced probability divergences, which have been increasingly popular in machine learning applications. We derived theoretical results about their induced topology as well as their statistical efficiency in terms of number of samples and projections, and we empirically illustrated our findings on different setups. Specifically, we proved that the preserved topology and dimension-free sample complexity are intrinsic to slicing. Since this was unclear in the previous literature, which combined slicing with a specific distance, our unified treatment of these results brings insight to the properties of particular instances used in practice. The gains in statistical efficiency could be explained by an ability of slicing to overlook irrelevant characteristics of the distributions. An important question for future work is then to understand precisely what geometrical features are well preserved by the slicing operation. Another interesting future direction is to extend our analysis to the recently proposed 'max-sliced' [12] and 'generalized' sliced divergences [15].

## Broader Impact

This paper is focused on the theoretical properties of sliced probability divergences, which have become increasingly popular in recent years due to their applications on implicit generative modeling. Our analysis uncovers the topological and statistical consequences of the slicing operation, and aims at providing answers to the question *"When and why do sliced divergences perform well in practice?"*. We believe that our theory would provide useful guidelines for practitioners working in this field, in terms of designing new sliced divergences as well as obtaining a better understanding on the existing sliced divergences. Our contributions are mainly theoretical, and we believe these will not pose any negative or positive ethical or societal consequence in the broad sense.

## Acknowledgments and Disclosure of Funding

This work is partly supported by the French National Research Agency (ANR) as a part of the FBIMATRIX project (ANR-16-CE23-0014) and by the industrial chair Machine Learning for Big Data from Télécom Paris. Alain Durmus acknowledges support from the Polish National Science Center grant (NCN UMO-2018/31/B/ST1/00253). The authors are grateful to Christos Tsirigotis for the fruitful discussion which motivated this work, and to an anonymous reviewer who gave an argument to improve the initial bound in Proposition 2.

## Footnotes

*Corresponding author: `kimia.nadjahi@telecom-paris.fr`

[2]See https://github.com/kimiandj/sliced_div

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
