[Supplementary Material]

# Statistical and Topological Properties of Sliced Probability Divergences
## SUPPLEMENTARY DOCUMENT

**Kimia Nadjahi**[1*], **Alain Durmus**[2], **Lénaïc Chizat**[3],
**Soheil Kolouri**[4], **Shahin Shahrampour**[5], **Umut Şimşekli**[1,6]

1: LTCI, Télécom Paris, Institut Polytechnique de Paris, France
2: Centre Borelli, ENS Paris-Saclay, CNRS, Université Paris-Saclay, France
3: Laboratoire de Mathématiques d'Orsay, CNRS, Université Paris-Saclay, France
4: HRL Laboratories, LLC., Malibu, CA, USA
5: Texas A&M University, College Station, TX, USA
6: Department of Statistics, University of Oxford, UK

## S1    Postponed proofs for Section 3

### S1.1    Proof of Proposition 1

*Proof of Proposition 1.* (i) The fact that $\mathbf{S\Delta}_p$ is non-negative (or symmetric) if $\mathbf{\Delta}$ is, immediately follows from the definition of $\mathbf{S\Delta}_p$ (4).

(ii) Assume that $\mathbf{\Delta}$ satisfies the identity of indiscernibles, *i.e.* for $\mu', \nu' \in \mathcal{P}(\mathbb{R})$, $\mathbf{\Delta}(\mu', \nu') = 0$ if and only if $\mu' = \nu'$. For any $\mu \in \mathcal{P}(\mathbb{R}^d)$ and $\theta \in \mathbb{S}^{d-1}$, $\mathbf{\Delta}(\theta_\sharp^\star \mu, \theta_\sharp^\star \mu) = 0$, therefore $\mathbf{S\Delta}_p(\mu, \mu) = 0$ by its definition (4). Now, consider $\mu, \nu \in \mathcal{P}(\mathbb{R}^d)$ such that $\mathbf{S\Delta}_p(\mu, \nu) = 0$. Then, by the definition of $\mathbf{S\Delta}_p$ (4), we have $\mathbf{\Delta}(\theta_\sharp^\star \mu, \theta_\sharp^\star \nu) = 0$ for $\boldsymbol{\sigma}$-almost every ($\boldsymbol{\sigma}$-a.e.) $\theta \in \mathbb{S}^{d-1}$, therefore $\theta_\sharp^\star \mu = \theta_\sharp^\star \nu$ for $\boldsymbol{\sigma}$-a.e. $\theta \in \mathbb{S}^{d-1}$. Next, we use the same technique as in [1, Proposition 5.1.2]: for any measure $\xi \in \mathcal{P}(\mathbb{R}^s)$ ($s \geq 1$), $\mathcal{F}[\xi]$ denotes the Fourier transform of $\xi$ and is defined as, for any $w \in \mathbb{R}^s$,

$$\mathcal{F}[\xi](w) = \int_{\mathbb{R}^s} e^{-\mathrm{i}\langle w, x\rangle} \mathrm{d}\xi(x) \ .$$

Then, by using (S1) and the property of pushforward measures, we have for any $t \in \mathbb{R}$ and $\theta \in \mathbb{S}^{d-1}$,

$$\mathcal{F}[\theta_\sharp^\star \mu](t) = \int_{\mathbb{R}} e^{-\mathrm{i}tu} \mathrm{d}\theta_\sharp^\star \mu(u) = \int_{\mathbb{R}^d} e^{-\mathrm{i}t\langle \theta, x\rangle} \mathrm{d}\mu(x) = \mathcal{F}[\mu](t\theta) \ . \tag{S1}$$

Since for $\boldsymbol{\sigma}$-a.e. $\theta \in \mathbb{S}^{d-1}$, $\theta_\sharp^\star \mu = \theta_\sharp^\star \nu$ thus $\mathcal{F}[\theta_\sharp^\star \mu] = \mathcal{F}[\theta_\sharp^\star \nu]$, we obtain $\mathcal{F}[\mu] = \mathcal{F}[\nu]$. By the injectivity of the Fourier transform, we conclude that $\mu = \nu$.

(iii) Suppose $\mathbf{\Delta}$ is a metric. Based on the previous results, to show that $\mathbf{S\Delta}_p$ is a metric, all we need to prove here is that it verifies the triangle inequality. Let $\mu, \nu, \xi \in \mathcal{P}(\mathbb{R}^d)$. Using that $\mathbf{\Delta}$ satisfies the triangle inequality and the Minkowski inequality in $\mathrm{L}^p(\mathbb{S}^{d-1}, \boldsymbol{\sigma})$, we get

$$
\begin{aligned}
\mathbf{S\Delta}_p(\mu, \nu) &= \left\{ \int_{\mathbb{S}^{d-1}} \mathbf{\Delta}^p(\theta_\sharp^\star \mu, \theta_\sharp^\star \nu) \mathrm{d}\boldsymbol{\sigma}(\theta) \right\}^{1/p} \\
&\leq \left\{ \int_{\mathbb{S}^{d-1}} \left[ \mathbf{\Delta}(\theta_\sharp^\star \mu, \theta_\sharp^\star \xi) + \mathbf{\Delta}(\theta_\sharp^\star \xi, \theta_\sharp^\star \nu) \right]^p \mathrm{d}\boldsymbol{\sigma}(\theta) \right\}^{1/p} \\
&\leq \left\{ \int_{\mathbb{S}^{d-1}} \mathbf{\Delta}^p(\theta_\sharp^\star \mu, \theta_\sharp^\star \xi) \mathrm{d}\boldsymbol{\sigma}(\theta) \right\}^{1/p} + \left\{ \int_{\mathbb{S}^{d-1}} \mathbf{\Delta}^p(\theta_\sharp^\star \xi, \theta_\sharp^\star \nu) \mathrm{d}\boldsymbol{\sigma}(\theta) \right\}^{1/p} \\
&\leq \mathbf{S\Delta}_p(\mu, \xi) + \mathbf{S\Delta}_p(\xi, \nu) \ .
\end{aligned}
$$

$\square$

## S1.2   Proof of Theorem 1

We start by proving Lemma S1 below, which extends [2, Lemma S13] to the more general class of Sliced Probability Divergences.

**Lemma S1.** *Consider $(\mu_k)_{k\in\mathbb{N}}$ a sequence in $\mathcal{P}(\mathbb{R}^d)$ satisfying $\lim_{k\to\infty}\mathbf{S\Delta}_1(\mu_k,\mu)=0$, with $\mu\in\mathcal{P}(\mathbb{R}^d)$, and assume that the convergence in $\mathbf{\Delta}$ implies the weak convergence in $\mathcal{P}(\mathbb{R})$. Then, there exists an increasing function $\phi:\mathbb{N}\to\mathbb{N}$ such that the subsequence $(\mu_{\phi(k)})_{k\in\mathbb{N}}$ converges weakly to $\mu$.*

*Proof.* We assume that $\lim_{k\to\infty}\mathbf{S\Delta}_1(\mu_k,\mu)=0$, *i.e.*:

$$\lim_{k\to\infty}\int_{\mathbb{S}^{d-1}}\mathbf{\Delta}(\theta_\sharp^\star\mu_k,\theta_\sharp^\star\mu)\mathrm{d}\boldsymbol{\sigma}(\theta)=0 \tag{S2}$$

By [3, Theorem 2.2.5], (S2) implies that, there exists an increasing function $\phi:\mathbb{N}\to\mathbb{N}$ such that for $\boldsymbol{\sigma}$-a.e. $\theta\in\mathbb{S}^{d-1}$, $\lim_{k\to\infty}\mathbf{\Delta}(\theta_\sharp^\star\mu_{\phi(k)},\theta_\sharp^\star\mu)=0$. Since $\mathbf{\Delta}$ is assumed to imply weak convergence in $\mathcal{P}(\mathbb{R})$, then, for $\boldsymbol{\sigma}$-a.e. $\theta\in\mathbb{S}^{d-1}$, $(\theta_\sharp^\star\mu_{\phi(k)})_{k\in\mathbb{N}}$ converges weakly to $\theta_\sharp^\star\mu$. By Lévy's characterization [4, Theorem 4.3], we have for $\boldsymbol{\sigma}$-a.e. $\theta\in\mathbb{S}^{d-1}$ and any $s\in\mathbb{R}$,

$$\lim_{k\to\infty}\Phi_{\theta_\sharp^\star\mu_{\phi(k)}}(s)=\Phi_{\theta_\sharp^\star\mu}(s)\ ,$$

where $\Phi_\nu$ is the characteristic function of $\nu\in\mathcal{P}(\mathbb{R}^s)$ ($s\geq 1$) and is defined as: for any $v\in\mathbb{R}^s$, $\Phi_\nu(v)=\int_{\mathbb{R}^s}\mathrm{e}^{\mathrm{i}\langle v,w\rangle}\mathrm{d}\nu(w)$. Therefore, for Lebesgue (Leb)-almost every $z\in\mathbb{R}^d$,

$$\lim_{k\to\infty}\Phi_{\mu_{\phi(k)}}(z)=\Phi_\mu(z)\ . \tag{S3}$$

We now use (S3) to show that $(\mu_{\phi(k)})_{k\in\mathbb{N}}$ converges weakly to $\mu$. By [5, Problem 1.11, Chapter 1], this boils down to proving that, for any $f:\mathbb{R}^d\to\mathbb{R}$ continuous with compact support,

$$\lim_{k\to\infty}\int_{\mathbb{R}^d}f(z)\mathrm{d}\mu_{\phi(k)}(z)=\int_{\mathbb{R}^d}f(z)\mathrm{d}\mu(z)\ . \tag{S4}$$

Consider $\sigma>0$ and a continuous function $f:\mathbb{R}^d\to\mathbb{R}$ with compact support. We introduce the function $f_\sigma$ defined as: for any $x\in\mathbb{R}^d$,

$$f_\sigma(x)=(2\pi\sigma^2)^{-d/2}\int_{\mathbb{R}^d}f(x-z)\exp\left(-\|z\|^2/(2\sigma^2)\right)\mathrm{d}z=f*g_\sigma(x)\ ,$$

where $*$ denotes the convolution product, and $g_\sigma$ is the density of the $d$-dimensional Gaussian with zero mean and covariance matrix $\sigma^2\mathbf{I}_d$. First, we prove that (S4) holds with $f_\sigma$ in place of $f$. The characteristic function associated to a $d$-dimensional Gaussian random variable $G$ with zero mean and covariance matrix $(1/\sigma^2)\mathbf{I}_d$ is given by: for any $z\in\mathbb{R}^d$, $\mathbb{E}\left[\mathrm{e}^{\mathrm{i}\langle z,G\rangle}\right]=\mathrm{e}^{-\|z\|^2/(2\sigma^2)}$. By plugging this in the definition of $f_\sigma$ and using Fubini's theorem, we obtain for any $k\in\mathbb{N}$,

$$\int_{\mathbb{R}^d}f_\sigma(z)\mathrm{d}\mu_{\phi(k)}(z)=\int_{\mathbb{R}^d}\int_{\mathbb{R}^d}f(w)g_\sigma(z-w)\mathrm{d}w\mathrm{d}\mu_{\phi(k)}(z)$$

$$=(2\pi\sigma^2)^{-d/2}\int_{\mathbb{R}^d}\int_{\mathbb{R}^d}f(w)\int_{\mathbb{R}^d}\mathrm{e}^{\mathrm{i}\langle z-w,x\rangle}g_{1/\sigma}(x)\mathrm{d}x\mathrm{d}w\mathrm{d}\mu_{\phi(k)}(z)$$

$$=(2\pi\sigma^2)^{-d/2}\int_{\mathbb{R}^d}\int_{\mathbb{R}^d}f(w)\mathrm{e}^{-\mathrm{i}\langle w,x\rangle}g_{1/\sigma}(x)\Phi_{\mu_{\phi(k)}}(x)\mathrm{d}x\mathrm{d}w$$

$$=(2\pi\sigma^2)^{-d/2}\int_{\mathbb{R}^d}\mathcal{F}[f](x)g_{1/\sigma}(x)\Phi_{\mu_{\phi(k)}}(x)\mathrm{d}x\ , \tag{S5}$$

where $\mathcal{F}[f](x)=\int_{\mathbb{R}^d}f(w)\mathrm{e}^{-\mathrm{i}\langle w,x\rangle}\mathrm{d}w$ is the Fourier transform of $f$. Since the support of $f$ is assumed to be compact, $\mathcal{F}[f]$ exists and is bounded by $\int_{\mathbb{R}^d}|f(w)|\,\mathrm{d}w<+\infty$, therefore, for any $k\in\mathbb{N}$ and $x\in\mathbb{R}^d$,

$$\left|\mathcal{F}[f](x)g_{1/\sigma}(x)\Phi_{\mu_{\phi(k)}}(x)\right|\leq g_{1/\sigma}(x)\int_{\mathbb{R}^d}|f(w)|\mathrm{d}w\ .$$

We can prove with similar techniques that (S5) holds with $\mu$ in place of $\mu_{\phi(k)}$, *i.e.*:

$$\int_{\mathbb{R}^d} f_\sigma(z) \mathrm{d}\mu(z) = (2\pi\sigma^2)^{-d/2} \int_{\mathbb{R}^d} \mathcal{F}[f](x) g_{1/\sigma}(x) \Phi_\mu(x) \mathrm{d}x . \tag{S6}$$

Using (S3), (S5), (S6) and Lebesgue's Dominated Convergence Theorem, we obtain:

$$\lim_{k\to\infty} (2\pi\sigma^2)^{-d/2} \int_{\mathbb{R}^d} \mathcal{F}[f](x) g_{1/\sigma}(x) \Phi_{\mu_{\phi(k)}}(x) \mathrm{d}x = (2\pi\sigma^2)^{-d/2} \int_{\mathbb{R}^d} \mathcal{F}[f](x) g_{1/\sigma}(x) \Phi_\mu(x) \mathrm{d}x ,$$

$$\textit{i.e.,} \quad \lim_{k\to\infty} \int_{\mathbb{R}^d} f_\sigma(z) \mathrm{d}\mu_{\phi(k)}(z) = \int_{\mathbb{R}^d} f_\sigma(z) \mathrm{d}\mu(z) . \tag{S7}$$

We can now prove (S4): for any $\sigma > 0$,

$$\left| \int_{\mathbb{R}^d} f(z) \mathrm{d}\mu_{\phi(k)}(z) - \int_{\mathbb{R}^d} f(z) \mathrm{d}\mu(z) \right|$$

$$\leq 2 \sup_{z\in\mathbb{R}^d} |f(z) - f_\sigma(z)| + \left| \int_{\mathbb{R}^d} f_\sigma(z) \mathrm{d}\mu_{\phi(k)}(z) - \int_{\mathbb{R}^d} f_\sigma(z) \mathrm{d}\mu(z) \right| .$$

By (S7), we deduce that for any $\sigma > 0$,

$$\limsup_{k\to+\infty} \left| \int_{\mathbb{R}^d} f(z) \mathrm{d}\mu_{\phi(k)}(z) - \int_{\mathbb{R}^d} f(z) \mathrm{d}\mu(z) \right| \leq 2 \sup_{z\in\mathbb{R}^d} |f(z) - f_\sigma(z)| ,$$

and since $\lim_{\sigma\to 0} \sup_{z\in\mathbb{R}^d} |f(z) - f_\sigma(z)| = 0$ [6, Theorem 8.14-b], we conclude that $(\mu_{\phi(k)})_{k\in\mathbb{N}}$ converges weakly to $\mu$.

$\square$

We can now prove Theorem 1.

*Proof of Theorem 1.* Let $p \in [1, \infty)$ and $(\mu_k)_{k\in\mathbb{N}}$ be a sequence of probability measures in $\mathcal{P}(\mathbb{R}^d)$.

First, suppose $(\mu_k)_{k\in\mathbb{N}}$ converges weakly to $\mu \in \mathcal{P}(\mathbb{R}^d)$. By the continuous mapping theorem, since for any $\theta \in \mathbb{S}^{d-1}$, $\theta^\star$ is a bounded linear form thus continuous, then $(\theta^\star_\sharp \mu_k)_{k\in\mathbb{N}}$ converges weakly to $\theta^\star_\sharp \mu$. Therefore, according to our assumption on $\boldsymbol{\Delta}$, for any $\theta \in \mathbb{S}^{d-1}$,

$$\lim_{k\to\infty} \boldsymbol{\Delta}(\theta^\star_\sharp \mu_k, \theta^\star_\sharp \mu) = 0 . \tag{S8}$$

Besides, $\boldsymbol{\Delta}$ is assumed to be non-negative and bounded. Hence, there exists $M > 0$ such that, for any $k \in \mathbb{N}$,

$$\boldsymbol{\Delta}^p(\theta^\star_\sharp \mu_k, \theta^\star_\sharp \mu) \leq M . \tag{S9}$$

Using (S8), (S9) and the bounded convergence theorem, we obtain

$$\lim_{k\to\infty} \mathbf{S\Delta}_p^p(\mu_k, \mu) = \lim_{k\to\infty} \int_{\mathbb{S}^{d-1}} \boldsymbol{\Delta}^p(\theta^\star_\sharp \mu_k, \theta^\star_\sharp \mu) \mathrm{d}\boldsymbol{\sigma}(\theta) = \int_{\mathbb{S}^{d-1}} 0^p \, \mathrm{d}\boldsymbol{\sigma}(\theta) = 0 . \tag{S10}$$

Since the mapping $t \mapsto t^{1/p}$ is continuous on $\mathbb{R}+$ (and can be applied to $\mathbf{S\Delta}_p^p$, which is non-negative by the non-negativity of $\boldsymbol{\Delta}$ and Proposition 1), then (S10) implies $\lim_{k\to\infty} \mathbf{S\Delta}_p(\mu_k, \mu) = 0$.

Now, let us prove the other implication, *i.e.* $\lim_{k\to\infty} \mathbf{S\Delta}_p(\mu_k, \mu) = 0$ implies the weak convergence of $(\mu_k)_{k\in\mathbb{N}}$ to $\mu$, given the assumptions on $\boldsymbol{\Delta}$. This result is a generalization of [2, Theorem 1], and is proved analogously, using Lemma S1: consider $(\mu_k)_{k\in\mathbb{N}}$ and $\mu$ in $\mathcal{P}(\mathbb{R}^d)$ such that

$$\lim_{k\to\infty} \mathbf{S\Delta}_p(\mu_k, \mu) = 0 , \tag{S11}$$

and suppose $(\mu_k)_{k\in\mathbb{N}}$ does not converge weakly to $\mu$. Therefore, $\lim_{k\to\infty} \mathbf{d}_\mathcal{P}(\mu_k, \mu) \neq 0$, where $\mathbf{d}_\mathcal{P}$ is the Lévy-Prokhorov metric, *i.e.* there exists $\epsilon > 0$ and a subsequence $(\mu_{\psi(k)})_{k\in\mathbb{N}}$ with $\psi : \mathbb{N} \to \mathbb{N}$ increasing, such that for any $k \in \mathbb{N}$,

$$\mathbf{d}_\mathcal{P}(\mu_{\psi(k)}, \mu) > \epsilon . \tag{S12}$$

On the other hand, an application of Hölder's inequality on $\mathbb{S}^{d-1}$ gives for any $\mu, \nu$ in $\mathcal{P}(\mathbb{R}^d)$,

$$\mathbf{S\Delta}_1(\mu, \nu) \leq \mathbf{S\Delta}_p(\mu, \nu) .$$

Then, by (S11), $\lim_{k \to \infty} \mathbf{S\Delta}_1(\mu_{\psi(k)}, \mu) = 0$. Since we assume the convergence in $\mathbf{\Delta}$ implies the weak convergence in $\mathcal{P}(\mathbb{R})$, Lemma S1 gives us: there exists a subsequence $(\mu_{\phi(\psi(k))})_{k \in \mathbb{N}}$ with $\phi : \mathbb{N} \to \mathbb{N}$ increasing such that $(\mu_{\phi(\psi(k))})_{k \in \mathbb{N}}$ converges weakly to $\mu$. This is equivalent to $\lim_{k \to \infty} \mathbf{d}_\mathcal{P}(\mu_{\phi(\psi(k))}, \mu) = 0$, which contradicts (S12). We conclude that (S11) implies the weak convergence of $(\mu_k)_{k \in \mathbb{N}}$ to $\mu$.

$\square$

### S1.3 Proof of Theorem 2

*Proof of Theorem 2.* Let $p \in [1, \infty)$ and $\mu, \nu \in \mathcal{P}(\mathbb{R}^d)$.

$$(\mathbf{S\gamma}_{\widetilde{\mathsf{F}},p})^p(\mu, \nu) = \int_{\mathbb{S}^{d-1}} \boldsymbol{\gamma}_{\widetilde{\mathsf{F}}}^p(\theta_\sharp^\star \mu, \theta_\sharp^\star \nu) \mathrm{d}\boldsymbol{\sigma}(\theta)$$

$$= \int_{\mathbb{S}^{d-1}} \left\{ \sup_{\tilde{f} \in \widetilde{\mathsf{F}}} \left| \int_{\mathbb{R}} \tilde{f}(t) \, \mathrm{d}(\theta_\sharp^\star \mu - \theta_\sharp^\star \nu)(t) \right| \right\}^p \mathrm{d}\boldsymbol{\sigma}(\theta)$$

$$= \int_{\mathbb{S}^{d-1}} \left| \int_{\mathbb{R}} \tilde{f}^*(t) \mathrm{d}(\theta_\sharp^\star \mu - \theta_\sharp^\star \nu)(t) \right|^p \mathrm{d}\boldsymbol{\sigma}(\theta)$$

$$= \int_{\mathbb{S}^{d-1}} \left| \int_{\mathbb{R}^d} \tilde{f}^*\big(\theta^\star(x)\big) \mathrm{d}(\mu - \nu)(x) \right|^p \mathrm{d}\boldsymbol{\sigma}(\theta) , \qquad \text{(S13)}$$

with $\tilde{f}^* = \mathrm{argmax}_{\tilde{f} \in \widetilde{\mathsf{F}}} \left| \int_{\mathbb{R}} \tilde{f}(t) \mathrm{d}\theta_\sharp^\star \mu(t) - \int_{\mathbb{R}} \tilde{f}(t) \mathrm{d}\theta_\sharp^\star \nu(t) \right|$, which is assumed to exist. Note that (S13) results from applying the property of pushforward measures.

By definition of $\mathsf{F}$, for any $\theta \in \mathbb{S}^{d-1}$, there exists $f_\theta^* \in \mathsf{F}$ such that $f_\theta^* = \tilde{f}^* \circ \theta^\star$. Therefore, we obtain

$$(\mathbf{S\gamma}_{\mathsf{F},p})^p(\mu, \nu) = \int_{\mathbb{S}^{d-1}} \left| \int_{\mathbb{R}^d} f_\theta^*(x) \mathrm{d}(\mu - \nu)(x) \right|^p \mathrm{d}\boldsymbol{\sigma}(\theta)$$

$$\leq \int_{\mathbb{S}^{d-1}} \left\{ \sup_{f \in \mathsf{F}} \left| \int_{\mathbb{R}^d} f(x) \mathrm{d}(\mu - \nu)(x) \right| \right\}^p \mathrm{d}\boldsymbol{\sigma}(\theta)$$

$$= \boldsymbol{\gamma}_{\mathsf{F}}^p(\mu, \nu) \int_{\mathbb{S}^{d-1}} \mathrm{d}\boldsymbol{\sigma}(\theta) = \boldsymbol{\gamma}_{\mathsf{F}}^p(\mu, \nu) ,$$

which completes the proof.

$\square$

Informally, the condition on the function classes in Theorem 2 requires that $\mathsf{F}$ and $\widetilde{\mathsf{F}}$ should be linked to each other in the way that $\mathsf{F}$ should be large enough to contain the composition of *all* elements of $\widetilde{\mathsf{F}}$ with *all* possible linear forms $\theta^\star$ for $\theta \in \mathbb{S}^{d-1}$. Let us illustrate this condition by considering the Wasserstein distance of order 1. In this case, $\mathsf{F}$ is the set of 1-Lipschitz functions from $\mathbb{R}^d$ to $\mathbb{R}$, and $\widetilde{\mathsf{F}}$ is the set of 1-Lipschitz functions from $\mathbb{R}$ to $\mathbb{R}$. Then, the condition on $\mathsf{F}$ boils down to showing that the composition of any $\tilde{f} \in \widetilde{\mathsf{F}}$ with any linear projection $\theta^\star$ results in a 1-Lipschitz function in $\mathbb{R}^d$, which is simply true since $\tilde{f}$ is 1-Lipschitz and $\|\theta\| = 1$ for all $\theta \in \mathbb{S}^{d-1}$.

In the next three corollaries, we formally prove that Theorem 2 holds for the Wasserstein distance of order 1 $\mathbf{W}_1$, total variation distance $\mathbf{TV}$ and maximum mean discrepancy $\mathbf{MMD}$. We denote by $\mathbf{SW}_1$, $\mathbf{STV}_p$ and $\mathbf{SMMD}_p$ the respective sliced versions of these IPMs with order $p \in [1, \infty)$.

**Corollary S1.** *Let $p \in [1, \infty)$. For any $\mu, \nu \in \mathcal{P}_1(\mathbb{R}^d)$, $\mathbf{SW}_1(\mu, \nu) \leq \mathbf{W}_1(\mu, \nu)$.*

*Proof.* Choose $\widetilde{\mathsf{F}} = \{\tilde{f} : \mathbb{R} \to \mathbb{R} : \|\tilde{f}\|_{\mathrm{Lip}} \leq 1\}$, where $\|\tilde{f}\|_{\mathrm{Lip}} = \sup_{x,y \in \mathbb{R}^d, x \neq y} \{|\tilde{f}(x) - \tilde{f}(y)| / \|x - y\|\}$. Let $f : \mathbb{R}^d \to \mathbb{R}$ such that $f = \tilde{f} \circ \theta^\star$ with $\tilde{f} \in \widetilde{\mathsf{F}}, \theta \in \mathbb{S}^{d-1}$. Then, by using the

Cauchy-Schwarz inequality and the definition of $\widetilde{\mathsf{F}}$, we have for any $x, y \in \mathbb{R}^d$,

$$|f(x) - f(y)| = \left| \tilde{f}\left(\theta^\star(x)\right) - \tilde{f}\left(\theta^\star(y)\right) \right| \le \left| \langle \theta, x - y \rangle \right| \le \|\theta^\star\| \, \|x - y\| \le \|x - y\| \, .$$

Therefore, $f \in \mathsf{F} = \{f : \mathbb{R}^d \to \mathbb{R} \; : \; \|f\|_{\mathrm{Lip}} \le 1\}$. Corollary S1 follows from the application of Theorem 2 along with the definition of $\mathbf{W}_1$.

$\square$

Note that Corollary S1 is not a new result: the fact that $\mathbf{SW}_p$ is bounded above by $\mathbf{W}_p$ for $p \in [1, \infty)$ was established in [1, Proposition 5.1.3]. While their result is proved using the primal formulation of the OT problem, we used the dual formulation available for $p = 1$ to illustrate the applicability of Theorem 2. Our result is thus consistent with the existing results in the literature.

**Corollary S2.** *Let $p \in [1, \infty)$. For any $\mu, \nu \in \mathcal{P}(\mathbb{R}^d)$,*

$$\mathbf{STV}_p(\mu, \nu) \le \mathbf{TV}(\mu, \nu) \, .$$

*Proof.* Choose $\widetilde{\mathsf{F}} = \left\{ \tilde{f} : \mathbb{R} \to \mathbb{R}, \; \|\tilde{f}\|_\infty \le 1 \right\}$, and let $f : \mathbb{R}^d \to \mathbb{R}$ such that $f = \tilde{f} \circ \theta^\star$ with $\tilde{f} \in \widetilde{\mathsf{F}}, \theta \in \mathbb{S}^{d-1}$. Then,

$$\|f\|_\infty = \|\tilde{f} \circ \theta^\star\|_\infty = \sup_{x \in \mathbb{R}^d} \left| \tilde{f}\left(\theta^\star(x)\right) \right| \le \sup_{t \in \mathbb{R}} \left| \tilde{f}(t) \right| = \|\tilde{f}\|_\infty \le 1 \, ,$$

hence, $f \in \mathsf{F} = \left\{ f : \mathbb{R}^d \to \mathbb{R} \; : \; \|f\|_\infty \le 1 \right\}$. We obtain the final result by using Theorem 2 and the definition of TV.

$\square$

**Corollary S3.** *Let $\widetilde{\mathsf{F}} \subset \mathbb{M}_b(\mathbb{R})$ be the unit ball of the RKHS with reproducing kernel $\tilde{k}$, and $k$ be the positive definite kernel such that for any $x_i, x_j \in \mathbb{R}^d$,*

$$k(x_i, x_j) = \int_{\mathbb{S}^{d-1}} \tilde{k}\left(\theta^\star(x_i), \theta^\star(x_j)\right) \mathrm{d}\boldsymbol{\sigma}(\theta) \, .$$

*Define $\mathsf{F} \subset \mathbb{M}_b(\mathbb{R}^d)$ as the unit ball of the RKHS whose reproducing kernel $\hat{k}$ satisfies $k - \hat{k}$ is positive definite. Then, for any $p \in [1, \infty)$ and $\mu, \nu \in \mathcal{P}(\mathbb{R}^d)$,*

$$\mathbf{SMMD}_p(\mu, \nu; \widetilde{\mathsf{F}}) \le \mathbf{MMD}(\mu, \nu; \mathsf{F}) \, ,$$

*where $\mathbf{MMD}(\cdot, \cdot \, ; \, \mathsf{F}')$ and $\mathbf{SMMD}_p(\cdot, \cdot \, ; \, \mathsf{F}')$ respectively denote the MMD and the Sliced-MMD of order $p$ in the RKHS whose unit ball is $\mathsf{F}'$.*

*In particular, this property holds for*

(i) *Linear kernels: $\tilde{k}(t_i, t_j) = t_i t_j$ for $t_i, t_j \in \mathbb{R}$, and $\hat{k}(x_i, x_j) = x_i^\top x_j / d'$ for $x_i, x_j \in \mathbb{R}$ and $d' \ge d$.*

(ii) *Radial basis function (RBF) kernels: let $h \ge 0$, $\tilde{k}(t_i, t_j) = e^{-|t_i - t_j|^2 / h}$ for $t_i, t_j \in \mathbb{R}$, and $\hat{k}(x_i, x_j) = e^{-\|x_i - x_j\|^2 / h}$ for $x_i, x_j \in \mathbb{R}^d$.*

*Proof.* Define $\widetilde{\mathsf{F}}$ as the unit ball of an RKHS whose reproducing kernel is denoted by $\tilde{k}$. Then, any $\tilde{f} \in \widetilde{\mathsf{F}}$ satisfies

$$\|\tilde{f}\|_{\widetilde{\mathsf{F}}}^2 = \sum_{i=1}^n \sum_{j=1}^n \alpha_i \alpha_j \tilde{k}(t_i, t_j) \le 1, \tag{S14}$$

where $n \in \mathbb{N}^*$, $\alpha_1, \ldots, \alpha_n \in \mathbb{R}$ and $t_1, \ldots, t_n \in \mathbb{R}$.

Consider $f : \mathbb{R}^d \to \mathbb{R}$ such that $f = \tilde{f} \circ \theta^*$ with $\tilde{f} \in \widetilde{\mathsf{F}}$ and $\theta \in \mathbb{S}^{d-1}$. By (S14), we have

$$\sum_{i=1}^n \sum_{j=1}^n \alpha_i \alpha_j \tilde{k}\left(\theta^\star(x_i), \theta^\star(x_j)\right) \le 1 \tag{S15}$$

The integration of (S15) over $\mathbb{S}^{d-1}$ give us

$$\int_{\mathbb{S}^{d-1}} \sum_{i=1}^{n} \sum_{j=1}^{n} \alpha_i \alpha_j \tilde{k}\big(\theta^{\star}(x_i), \theta^{\star}(x_j)\big) \mathrm{d}\boldsymbol{\sigma}(\theta) \leq \int_{\mathbb{S}^{d-1}} 1 \, \mathrm{d}\boldsymbol{\sigma}(\theta)$$

$$i.e., \quad \sum_{i=1}^{n} \sum_{j=1}^{n} \alpha_i \alpha_j \int_{\mathbb{S}^{d-1}} \tilde{k}\big(\theta^{\star}(x_i), \theta^{\star}(x_j)\big) \mathrm{d}\boldsymbol{\sigma}(\theta) \leq 1 \ . \tag{S16}$$

Define $k : \mathbb{R}^d \times \mathbb{R}^d \to \mathbb{R}$ as $k(x_i, x_j) = \int_{\mathbb{S}^{d-1}} \tilde{k}\big(\theta^{\star}(x_i), \theta^{\star}(x_j)\big) \mathrm{d}\boldsymbol{\sigma}(\theta)$ for $x_i, x_j \in \mathbb{R}^d$. Since $\tilde{k}$ is positive definite, so is $k$. By the Moore-Aronszajn theorem, there exists a unique RKHS with reproducing kernel $k$. Therefore, (S16) means that $f$ is in the unit ball of the RKHS associated with $k$.

Additionally, consider a positive definite kernel $\hat{k} : \mathbb{R}^d \times \mathbb{R}^d \to \mathbb{R}$ such that $k - \hat{k}$ is positive definite on $\mathbb{R}^d$. In other words, the following holds for any $n \in \mathbb{N}$, $v_1, \ldots, v_n \in \mathbb{R}$ and $x_1, \ldots, x_n \in \mathbb{R}^d$,

$$\sum_{i=1}^{n} \sum_{j=1}^{n} v_i v_j \{k(x_i, x_j) - \hat{k}(x_i, x_j)\} \geq 0 \ .$$

Then, by (S16), we obtain $\sum_{i=1}^{n} \sum_{j=1}^{n} \alpha_i \alpha_j \hat{k}(x_i, x_j) \leq 1$.

Therefore, any $f$ defined as $f = \tilde{f} \circ \theta$ with $\tilde{f} \in \widetilde{\mathsf{F}}$ and $\theta \in \mathbb{S}^{d-1}$ is in the unit ball of the RKHS associated with $\hat{k}$, which we denote by $\mathsf{F}$. By using Theorem 2 and the definition of MMD, we obtain the desired result: for any $p \in [1, \infty)$ and $\mu, \nu \in \mathcal{P}(\mathbb{R}^d)$,

$$\mathbf{SMMD}_p(\mu, \nu; \widetilde{\mathsf{F}}) \leq \mathbf{MMD}(\mu, \nu; \mathsf{F}) \ . \tag{S17}$$

Next, we show that this result holds for two popular choices of kernels. First, we choose $\tilde{k}$ as the linear kernel: $\tilde{k}(t_i, t_j) = t_i t_j$ for $t_i, t_j \in \mathbb{R}$. Define $\hat{k}$ as a rescaled version of the linear kernel in $\mathbb{R}^d$: $\hat{k}(x_i, x_j) = x_i^\top x_j / d'$ for $x_i, x_j \in \mathbb{R}^d$ and $d' \geq d$. Then, for any $n \in \mathbb{N}$, $v_1, \ldots, v_n \in \mathbb{R}$ and $x_1, \ldots, x_n \in \mathbb{R}^d$,

$$\sum_{i=1}^{n} \sum_{j=1}^{n} v_i v_j \{k(x_i, x_j) - \hat{k}(x_i, x_j)\} = \sum_{i=1}^{n} \sum_{j=1}^{n} v_i v_j \Big\{ \int_{\mathbb{S}^{d-1}} \theta(x_i) \theta(x_j) \mathrm{d}\boldsymbol{\sigma}(\theta) - x_i^\top x_j / d' \Big\}$$

$$= \sum_{i=1}^{n} \sum_{j=1}^{n} v_i v_j \Big\{ x_i^\top \Big( \int_{\mathbb{S}^{d-1}} \theta \theta^\top \mathrm{d}\boldsymbol{\sigma}(\theta) \Big) x_j - x_i^\top x_j / d' \Big\}$$

$$= \sum_{i=1}^{n} \sum_{j=1}^{n} v_i v_j x_i^\top x_j \Big( 1/d - 1/d' \Big) \geq 0 \ , \tag{S18}$$

where (S18) results from $\sum_{i=1}^{n} \sum_{j=1}^{n} v_i v_j x_i^\top x_j \geq 0$ (the linear kernel is positive definite) and $d' \geq d$. We conclude that (S17) holds with $\widetilde{\mathsf{F}}$ defined as the unit ball of the RKHS associated with the linear kernel $\tilde{k}(t_i, t_j) = t_i t_j$ for $t_i, t_j \in \mathbb{R}$, and $\mathsf{F}$ being the unit ball of the RKHS associated with the rescaled linear kernel $\hat{k}(x_i, x_j) = x_i^\top x_j / d'$ for $x_i, x_j \in \mathbb{R}^d$ and $d' \geq d$.

We conclude that (S17) holds with $\widetilde{\mathsf{F}}$ defined as the unit ball of the RKHS associated with the linear kernel $\tilde{k}(t_i, t_j) = t_i t_j$ for $t_i, t_j \in \mathbb{R}$, and $\mathsf{F}$ being the unit ball of the RKHS associated with the rescaled linear kernel $\hat{k}(x_i, x_j) = x_i^\top x_j / d$ for $x_i, x_j \in \mathbb{R}^d$.

We focus now on RBF kernels: let $h \geq 0$ and choose $\tilde{k}(t_i, t_j) = e^{-|t_i - t_j|^2/h}$ for $t_i, t_j \in \mathbb{R}$, and $\hat{k}(x_i, x_j) = e^{-\|x_i - x_j\|^2/h}$ for $x_i, x_j \in \mathbb{R}^d$. We have for any $x_i, x_j \in \mathbb{R}^d$,

$$
\begin{aligned}
k(x_i, x_j) &= \int_{\mathbb{S}^{d-1}} \tilde{k}\big(\theta(x_i), \theta(x_j)\big) \mathrm{d}\boldsymbol{\sigma}(\theta) = \int_{\mathbb{S}^{d-1}} e^{-|\theta^\top x_i - \theta^\top x_j|^2/h} \, \mathrm{d}\boldsymbol{\sigma}(\theta) \\
&= \int_{\mathbb{S}^{d-1}} e^{-|\theta^\top(x_i - x_j)|^2/h} \, \mathrm{d}\boldsymbol{\sigma}(\theta) \\
&= \int_{\mathbb{S}^{d-1}} e^{(-\|x_i - x_j\|^2/h)(\theta^\top(x_i - x_j)/\|x_i - x_j\|)^2} \mathrm{d}\boldsymbol{\sigma}(\theta) \\
&= M\left(\frac{1}{2}, \frac{d}{2}, -\frac{\|x_i - x_j\|^2}{h}\right) ,
\end{aligned}
\tag{S19}
$$

where $M(a, c, \kappa)$ stands for the confluent hypergeometric function evaluated at $a, c, \kappa \in \mathbb{R}$, and appears in the normalizing constant of the multivariate Watson distribution: see [7, Section 2.3] for more details.

$M$ satisfies the following property

$$
M\left(\frac{1}{2}, \frac{d}{2}, -\frac{\|x_i - x_j\|^2}{h}\right) = e^{-\|x_i - x_j\|^2/h} \, M\left(\frac{d-1}{2}, \frac{d}{2}, \frac{\|x_i - x_j\|^2}{h}\right) .
\tag{S20}
$$

Since $\|x_i - x_j\|^2/h \geq 0$ and $\kappa \mapsto M(\cdot, \cdot, \kappa)$ is increasing, we have

$$
M\left(\frac{d-1}{2}, \frac{d}{2}, \frac{\|x_i - x_j\|^2}{h}\right) \geq M\left(\frac{d-1}{2}, \frac{d}{2}, 0\right) = M\left(\frac{1}{2}, \frac{d}{2}, 0\right) = 1 .
\tag{S21}
$$

Finally, by using (S19) and (S20), we obtain: for any $n \in \mathbb{N}$, $v_1, \ldots, v_n \in \mathbb{R}$ and $x_1, \ldots, x_n \in \mathbb{R}^d$,

$$
\begin{aligned}
\sum_{i=1}^n \sum_{j=1}^n v_i v_j \{k(x_i, x_j) - \hat{k}(x_i, x_j)\} &= \sum_{i=1}^n \sum_{j=1}^n v_i v_j \left[ M\left(\frac{1}{2}, \frac{d}{2}, -\frac{\|x_i - x_j\|^2}{h}\right) - e^{-\|x_i - x_j\|^2/h} \right] \\
&= \sum_{i=1}^n \sum_{j=1}^n v_i v_j e^{-\|x_i - x_j\|^2/h} \left[ M\left(\frac{d-1}{2}, \frac{d}{2}, \frac{\|x_i - x_j\|^2}{h}\right) - 1 \right] \\
&\geq 0 ,
\end{aligned}
$$

where the last line follows from (S21) and $\sum_{i=1}^n \sum_{j=1}^n v_i v_j e^{-\|x_i - x_j\|^2/h} \geq 0$ (RBF kernels are positive definite). We conclude that $k - \hat{k}$ is positive definite, hence (S17) holds for RBF kernels.

$\square$

## S1.4 Proof of Theorem 3

*Proof of Theorem 3.* We start by upper bounding the distance between two regularized measures. Denote by $\mathrm{supp}(\zeta)$ the support of the function $\zeta$. Let $\varphi : \mathbb{R} \to \mathbb{R}_+^*$ be a smooth and even function verifying $\mathrm{supp}(\varphi) \subset [-1, 1]$ and $\int_{\mathbb{R}} \varphi(t) \mathrm{dLeb}(t) = 1$. Define $\varphi_\lambda(x) = \lambda^{-d} \varphi(\|x\|/\lambda)/\mathcal{A}(\mathbb{S}^{d-1})$, with $\mathcal{A}(\mathbb{S}^{d-1})$ denoting the surface area of the $d$-dimensional unit sphere: $\mathcal{A}(\mathbb{S}^{d-1}) = 2\pi^{d/2}/\Gamma(d/2)$, where $\Gamma$ is the gamma function. Denote by $\mathcal{F}[f]$ the Fourier transform of any function $f$ defined on $\mathbb{R}^s$ ($s \geq 1$), given by: for any $x \in \mathbb{R}^s$, $\mathcal{F}[f](x) = \int_{\mathbb{R}^s} f(w) e^{-\mathrm{i}\langle w, x\rangle} \mathrm{d}w$. Let $g \in \mathsf{G}$. By the isometry properties of the Fourier transform and the definition of $\varphi_\lambda$, we have

$$
\int_{\mathbb{R}^d} g(x) \mathrm{d}(\mu_\lambda - \nu_\lambda)(x) = \int_{\mathbb{R}^d} \mathcal{F}[g](w) \{\mathcal{F}[\mu](w) - \mathcal{F}[\nu](w)\} \mathcal{F}[\varphi](\lambda w) \mathrm{d}w ,
$$

where $\mu_\lambda = \mu * \varphi_\lambda$ and $\nu_\lambda = \nu * \varphi_\lambda$. By representing $w$ with its polar coordinates $(r, \theta) \in [0, \infty) \times \mathbb{S}^{d-1}$, we obtain

$$
\int_{\mathbb{R}^d} g(x) \mathrm{d}(\mu_\lambda - \nu_\lambda)(x) = \int_{\mathbb{S}^{d-1}} \int_0^\infty \mathcal{F}[g](r\theta) \{\mathcal{F}[\mu](r\theta) - \mathcal{F}[\nu](r\theta)\} \mathcal{F}[\varphi](\lambda r) r^{d-1} \mathrm{d}r \mathrm{d}\boldsymbol{\sigma}(\theta) .
$$

Since $g$ is a real function, $\mathcal{F}[g]$ is an even function, hence

$$\int_{\mathbb{R}^d} g(x)\mathrm{d}(\mu_\lambda - \nu_\lambda)(x)$$

$$= \frac{1}{2}\int_{\mathbb{S}^{d-1}}\int_{\mathbb{R}}\mathcal{F}[g](r\theta)\left\{\mathcal{F}[\mu](r\theta) - \mathcal{F}[\nu](r\theta)\right\}\mathcal{F}[\varphi](\lambda r)\,|r|^{d-1}\,\mathrm{d}r\mathrm{d}\boldsymbol{\sigma}(\theta)$$

$$= \frac{1}{2}\int_{\mathbb{S}^{d-1}}\int_{\mathbb{R}}\mathcal{F}[g](r\theta)\left\{\mathcal{F}[\theta_\sharp^\star\mu](r) - \mathcal{F}[\theta_\sharp^\star\nu](r)\right\}\mathcal{F}[\varphi](\lambda r)\,|r|^{d-1}\,\mathrm{d}r\mathrm{d}\boldsymbol{\sigma}(\theta) \tag{S22}$$

$$= \frac{1}{2}\int_{\mathbb{S}^{d-1}}\int_{\mathbb{R}}\int_{-R}^{R}\mathcal{F}[g](r\theta)e^{-\mathrm{i}ru}\mathrm{d}(\theta_\sharp^\star\mu - \theta_\sharp^\star\nu)(u)\mathcal{F}[\varphi](\lambda r)\,|r|^{d-1}\,\mathrm{d}r\mathrm{d}\boldsymbol{\sigma}(\theta) \tag{S23}$$

$$= \frac{1}{2}\int_{\mathbb{S}^{d-1}}\int_{\mathbb{R}}\int_{\mathbb{R}^d}\int_{-R}^{R} g(x)e^{-\mathrm{i}r(u+\langle\theta,x\rangle)}\left\{\mathrm{d}(\theta_\sharp^\star\mu - \theta_\sharp^\star\nu)(u)\right\}\mathcal{F}[\varphi](\lambda r)\,|r|^{d-1}\,\mathrm{d}x\mathrm{d}r\mathrm{d}\boldsymbol{\sigma}(\theta)\,,$$

where (S22) follows from (S1), (S23) results from the definition of the Fourier transform and the fact that $u \in [-R, R]$, and in the last line, we used the definition of the Fourier transform and Fubini's theorem. By making the change of variables $x \to x - u\theta$, we obtain

$$\int_{\mathbb{R}^d} g(x)\mathrm{d}(\mu_\lambda - \nu_\lambda)(x)$$

$$= \frac{1}{2}\int_{\mathbb{S}^{d-1}}\int_{\mathbb{R}}\int_{\mathbb{R}^d}\int_{-R}^{R} g(x - u\theta)e^{-\mathrm{i}r\langle\theta,x\rangle}\mathrm{d}(\theta_\sharp^\star\mu - \theta_\sharp^\star\nu)(u)\mathcal{F}[\varphi](\lambda r)\,|r|^{d-1}\,\mathrm{d}x\mathrm{d}r\mathrm{d}\boldsymbol{\sigma}(\theta)\,.$$

Since we assumed $\mathrm{supp}(\mu)$, $\mathrm{supp}(\nu)$ are included in $B_d(\mathbf{0}, R)$, then $\mathrm{supp}(\mu_\lambda)$, $\mathrm{supp}(\mu_\lambda)$ are in $B_d(\mathbf{0}, R + \lambda)$, and the domain of $x \mapsto g(x - u\theta)$ must be contained in $B_d(\mathbf{0}, 2R + \lambda)$. By Fubini's theorem and the definition of $\widetilde{\mathsf{G}}$, we have

$$\left|\int_{\mathbb{R}^d} g(x)\mathrm{d}(\mu_\lambda - \nu_\lambda)(x)\right|$$

$$\leq \frac{1}{2}\int_{\mathbb{R}}\int_{B_d(\mathbf{0},2R+\lambda)}\int_{\mathbb{S}^{d-1}}\left|\int_{-R}^{R} g(x - u\theta)\mathrm{d}(\theta_\sharp^\star\mu - \theta_\sharp^\star\nu)(u)e^{-\mathrm{i}r\langle\theta,x\rangle}\mathcal{F}[\varphi](\lambda r)\,|r|^{d-1}\right|\mathrm{d}\boldsymbol{\sigma}(\theta)\mathrm{d}x\mathrm{d}r$$

$$\leq \frac{1}{2}\int_{\mathbb{R}}\int_{B_d(\mathbf{0},2R+\lambda)}\int_{\mathbb{S}^{d-1}}\boldsymbol{\gamma}_{\widetilde{\mathsf{G}}}(\theta_\sharp^\star\mu, \theta_\sharp^\star\nu)\left|e^{-\mathrm{i}r\langle\theta,x\rangle}\mathcal{F}[\varphi](\lambda r)\,|r|^{d-1}\right|\mathrm{d}\boldsymbol{\sigma}(\theta)\mathrm{d}x\mathrm{d}r$$

$$\leq C(2R + \lambda)^d \int_{\mathbb{S}^{d-1}}\boldsymbol{\gamma}_{\widetilde{\mathsf{G}}}(\theta_\sharp^\star\mu, \theta_\sharp^\star\nu)\mathrm{d}\boldsymbol{\sigma}(\theta)\int_{\mathbb{R}}\lambda^{-d}\left|\mathcal{F}[\varphi](r)\,|r|^{d-1}\right|\mathrm{d}r \tag{S24}$$

$$\leq C(2R + \lambda)^d\lambda^{-d}\left(\int_{\mathbb{S}^{d-1}}\boldsymbol{\gamma}_{\widetilde{\mathsf{G}}}^p(\theta_\sharp^\star\mu, \theta_\sharp^\star\nu)\mathrm{d}\boldsymbol{\sigma}(\theta)\right)^{1/p}\int_{\mathbb{R}}\left|\mathcal{F}[\varphi](r)|r|^{d-1}\right|\mathrm{d}r \tag{S25}$$

$$\leq C_1(2R + \lambda)^d\lambda^{-d}\mathbf{S}\boldsymbol{\gamma}_{\widetilde{\mathsf{G}},p}(\mu, \nu)\,, \tag{S26}$$

where in (S24), $C > 0$ and does not depend on $\mu$ and $\nu$, (S25) results from applying Hölder's inequality on $\mathbb{S}^{d-1}$ if $p > 1$, and in (S26), $C_1 = C\int_{\mathbb{R}}\left|\mathcal{F}[\varphi](r)|r|^{d-1}\right|\mathrm{d}r$.

By using the definition of $\boldsymbol{\gamma}_{\mathsf{G}}$ and (S26), we obtain

$$\boldsymbol{\gamma}_{\mathsf{G}}(\mu_\lambda, \nu_\lambda) = \sup_{g\in\mathsf{G}}\left|\int_{\mathbb{R}^d} g(x)\mathrm{d}(\mu_\lambda - \nu_\lambda)(x)\right| \leq C_1(2R + \lambda)^d\lambda^{-d}\mathbf{S}\boldsymbol{\gamma}_{\widetilde{\mathsf{G}},p}(\mu, \nu)\,. \tag{S27}$$

We now relate $\boldsymbol{\gamma}_{\mathsf{G}}(\mu_\lambda, \nu_\lambda)$ with $\boldsymbol{\gamma}_{\mathsf{G}}(\mu, \nu)$. We start with the following estimate

$$\int_{\mathbb{R}^d} g(x)\mathrm{d}(\mu - \nu)(x) - \boldsymbol{\gamma}_{\mathsf{G}}(\mu_\lambda, \nu_\lambda)$$

$$\leq \int_{\mathbb{R}^d} g(x)\mathrm{d}(\mu - \nu)(x) - \int_{\mathbb{R}^d} g(x)\mathrm{d}(\mu_\lambda - \nu_\lambda)(x)$$

$$\leq \int_{\mathbb{R}^d}\left|g(x) - (\varphi_\lambda * g)(x)\right|\mathrm{d}\mu(x) + \int_{\mathbb{R}^d}\left|g(x) - (\varphi_\lambda * g)(x)\right|\mathrm{d}\nu(x) \tag{S28}$$

Since we assumed any $g \in \mathsf{G}$ is L-Lipschitz continuous, we can bound the integrand in (S28) as follows: for $x \in \mathbb{R}^d$,

$$
\begin{aligned}
\left| g(x) - (\varphi_\lambda * g)(x) \right| &= \left| \lambda^{-d} \int_{\mathbb{R}^d} \left( g(x) - g(y) \right) \varphi\big((x-y)/\lambda\big) \mathrm{d}y \right| \\
&\leq \lambda^{-d} \int_{\mathbb{R}^d} \left| g(x) - g(y) \right| \varphi\big((x-y)/\lambda\big) \mathrm{d}y \\
&\leq \mathrm{L}\lambda^{-d+1} \int_{\mathbb{R}^d} \|x-y\| \lambda^{-1} \varphi\big((x-y)/\lambda\big) \mathrm{d}y \\
&\leq \mathrm{L}\lambda^{-d+1} \int_{\mathbb{R}^d} \|u\| \lambda^{-1} \varphi\big(u/\lambda\big) \mathrm{d}u \leq \mathrm{L}\lambda \int \|z\| \varphi(z) \mathrm{d}z \ .
\end{aligned}
$$

Hence, by denoting by $M_1(\varphi)$ the moment of order 1 of $\varphi$, (S28) is bounded by

$$
\int_{\mathbb{R}^d} g(x)\mathrm{d}(\mu - \nu)(x) - \boldsymbol{\gamma}_{\mathsf{G}}(\mu_\lambda, \nu_\lambda) \leq 2\mathrm{L}M_1(\varphi)\lambda \ .
$$

Taking the supremum of both sides over $\mathsf{G}$ gives us

$$
\boldsymbol{\gamma}_{\mathsf{G}}(\mu, \nu) - \boldsymbol{\gamma}_{\mathsf{G}}(\mu_\lambda, \nu_\lambda) \leq 2\mathrm{L}M_1(\varphi)\lambda \ .
$$

By combining the above inequality with (S27), we get

$$
\begin{aligned}
\boldsymbol{\gamma}_{\mathsf{G}}(\mu, \nu) &\leq C_1(2R+\lambda)^d \lambda^{-d} \mathbf{S}\boldsymbol{\gamma}_{\widetilde{\mathsf{G}},p}(\mu, \nu) + 2\mathrm{L}M_1(\varphi)\lambda \\
&\leq C_2\lambda \Big( (2R+\lambda)^d \lambda^{-(d+1)} \mathbf{S}\boldsymbol{\gamma}_{\widetilde{\mathsf{G}},p}(\mu, \nu) + 1 \Big) \ ,
\end{aligned}
$$

with $C_2$ satisfying $C_2 \geq C_1$ and $C_2 \geq 2\mathrm{L}M_1(\varphi)$. Finally, by choosing $\lambda = R^{d/(d+1)} \mathbf{S}\boldsymbol{\gamma}_{\widetilde{\mathsf{G}},p}(\mu, \nu)^{1/(d+1)}$ and using the hypothesis that $\mathbf{S}\boldsymbol{\gamma}_{\widetilde{\mathsf{G}},p}$ is bounded, we obtain

$$
\begin{aligned}
\boldsymbol{\gamma}_{\mathsf{G}}(\mu, \nu) &\leq C_2 R^{d/(d+1)} \mathbf{S}\boldsymbol{\gamma}_{\widetilde{\mathsf{G}},p}(\mu, \nu)^{1/(d+1)} \Big( (2R+\lambda)^d R^{-d} + 1 \Big) \\
&\leq C_p \mathbf{S}\boldsymbol{\gamma}_{\widetilde{\mathsf{G}},p}(\mu, \nu)^{1/(d+1)},
\end{aligned}
$$

for some $C_p > 0$, as desired. This concludes the proof.

$\square$

As with Theorem 2, Theorem 3 assumes that the function classes $\mathsf{G}$ and $\widetilde{\mathsf{G}}$ are linked to each other and sufficiently regular. The condition on $\mathsf{G}$ is verified with $\mathbf{W}_1$ (simply by definition) and MMD (provided that the reproducing kernel is Lipschitz-continuous, which holds on compact spaces for classical choices of kernels), but not with TV. On the other hand, the second condition requires $\widetilde{\mathsf{G}}$ to be large enough to contain *any* possible slice $g(x - u\theta)$ for any $g \in \mathsf{G}$.

## S1.5 Proof of Corollary 1

*Proof of Corollary 1.* The desired result is obtained as a direct application of Theorems 2 and 3.

$\square$

## S1.6 Proof of Theorem 4

*Proof of Theorem 4.* Let $p \in [1, \infty)$ and $\mu, \nu$ in $\mathcal{P}(\mathbb{R}^d)$ with respective empirical measures $\hat{\mu}_n, \hat{\nu}_n$. By using the definition of $\mathbf{S}\boldsymbol{\Delta}_p$, the triangle inequality and the assumption on the sample complexity

of $\mathbf{\Delta}^p$, we have

$$
\begin{aligned}
\mathbb{E}\left|\mathbf{S\Delta}_p^p(\mu,\nu) - \mathbf{S\Delta}_p^p(\hat{\mu}_n,\hat{\nu}_n)\right| &= \mathbb{E}\left|\int_{\mathbb{S}^{d-1}}\left\{\mathbf{\Delta}^p(\theta_\sharp^\star\mu,\theta_\sharp^\star\nu) - \mathbf{\Delta}^p(\theta_\sharp^\star\hat{\mu}_n,\theta_\sharp^\star\hat{\nu}_n)\right\}\mathrm{d}\boldsymbol{\sigma}(\theta)\right| \\
&\leq \mathbb{E}\left\{\int_{\mathbb{S}^{d-1}}\left|\mathbf{\Delta}^p(\theta_\sharp^\star\mu,\theta_\sharp^\star\nu) - \mathbf{\Delta}^p(\theta_\sharp^\star\hat{\mu}_n,\theta_\sharp^\star\hat{\nu}_n)\right|\mathrm{d}\boldsymbol{\sigma}(\theta)\right\} \\
&\leq \int_{\mathbb{S}^{d-1}}\mathbb{E}\left|\mathbf{\Delta}^p(\theta_\sharp^\star\mu,\theta_\sharp^\star\nu) - \mathbf{\Delta}^p(\theta_\sharp^\star\hat{\mu}_n,\theta_\sharp^\star\hat{\nu}_n)\right|\mathrm{d}\boldsymbol{\sigma}(\theta) \\
&\leq \int_{\mathbb{S}^{d-1}}\beta(p,n)\mathrm{d}\boldsymbol{\sigma}(\theta) = \beta(p,n) ,
\end{aligned}
$$

which completes the proof.

$\square$

## S1.7 Proof of Theorem 5

*Proof of Theorem 5.* Let $p \in [1,\infty)$ and $\mu \in \mathcal{P}(\mathbb{R}^d)$ with corresponding empirical measure $\hat{\mu}_n$. By using the definition of $\mathbf{S\Delta}_p$, the triangle inequality and the assumed convergence rate of empirical measures in $\mathbf{\Delta}^p$, we obtain the convergence rate in $\mathbf{S\Delta}_p$ as follows

$$
\begin{aligned}
\mathbb{E}\left|\mathbf{S\Delta}_p^p(\hat{\mu}_n,\mu)\right| &= \mathbb{E}\left|\int_{\mathbb{S}^{d-1}}\mathbf{\Delta}^p(\theta_\sharp^\star\hat{\mu}_n,\theta_\sharp^\star\mu)\mathrm{d}\boldsymbol{\sigma}(\theta)\right| \leq \mathbb{E}\left\{\int_{\mathbb{S}^{d-1}}\left|\mathbf{\Delta}^p(\theta_\sharp^\star\hat{\mu}_n,\theta_\sharp^\star\mu)\right|\mathrm{d}\boldsymbol{\sigma}(\theta)\right\} \\
&\leq \int_{\mathbb{S}^{d-1}}\mathbb{E}\left|\mathbf{\Delta}^p(\theta_\sharp^\star\hat{\mu}_n,\theta_\sharp^\star\mu)\right|\mathrm{d}\boldsymbol{\sigma}(\theta) \leq \int_{\mathbb{S}^{d-1}}\alpha(p,n)\mathrm{d}\boldsymbol{\sigma}(\theta) = \alpha(p,n) . \quad \text{(S29)}
\end{aligned}
$$

Additionally, if we assume that $\mathbf{\Delta}$ satisfies non-negativity, symmetry and the triangle inequality, then $\mathbf{S\Delta}_p$ also verifies these three properties by Proposition 1, and we can derive its sample complexity: for any $\mu,\nu$ in $\mathcal{P}(\mathbb{R}^d)$ with respective empirical measures $\hat{\mu}_n,\hat{\nu}_n$, the triangle inequality give us

$$
\left|\mathbf{S\Delta}_p(\mu,\nu) - \mathbf{S\Delta}_p(\hat{\mu}_n,\hat{\nu}_n)\right| \leq \mathbf{S\Delta}_p(\hat{\mu}_n,\mu) + \mathbf{S\Delta}_p(\hat{\nu}_n,\nu) \quad \text{(S30)}
$$

By taking the expectation of (S30) with respect to $\hat{\mu}_n,\hat{\nu}_n$, we obtain

$$
\begin{aligned}
\mathbb{E}\left|\mathbf{S\Delta}_p(\mu,\nu) - \mathbf{S\Delta}_p(\hat{\mu}_n,\hat{\nu}_n)\right| &\leq \mathbb{E}\left|\mathbf{S\Delta}_p(\hat{\mu}_n,\mu)\right| + \mathbb{E}\left|\mathbf{S\Delta}_p(\hat{\nu}_n,\nu)\right| \\
&\leq \left\{\mathbb{E}\left|\mathbf{S\Delta}_p^p(\hat{\mu}_n,\mu)\right|\right\}^{1/p} + \left\{\mathbb{E}\left|\mathbf{S\Delta}_p^p(\hat{\nu}_n,\nu)\right|\right\}^{1/p} \quad \text{(S31)} \\
&\leq \alpha(p,n)^{1/p} + \alpha(p,n)^{1/p} = 2\alpha(p,n)^{1/p} , \quad \text{(S32)}
\end{aligned}
$$

where (S31) results from applying Hölder's inequality on $\mathbb{S}^{d-1}$ if $p > 1$, and (S32) follows from the convergence rate result in (S29).

$\square$

## S1.8 Proof of Theorem 6

*Proof of Theorem 6.* Let $p \in [1,\infty)$ and $\mu,\nu \in \mathcal{P}(\mathbb{R}^d)$. We recall that $\widehat{\mathbf{S\Delta}}_{p,L}(\mu,\nu)$ denotes the approximation of $\mathbf{S\Delta}_p(\mu,\nu)$ obtained with a Monte Carlo scheme that uniformly picks $L$ projection directions on $\mathbb{S}^{d-1}$ (cf. Equation (5) in the main document).

By using Hölder's inequality and the results on the moments of the Monte Carlo estimation error, we obtain

$$
\begin{aligned}
\mathbb{E}_{\theta\sim\boldsymbol{\sigma}}\left|\widehat{\mathbf{S\Delta}}_{p,L}^p(\mu,\nu) - \mathbf{S\Delta}_p^p(\mu,\nu)\right| &\leq \left\{\mathbb{E}_{\theta\sim\boldsymbol{\sigma}}\left|\widehat{\mathbf{S\Delta}}_{p,L}^p(\mu,\nu) - \mathbf{S\Delta}_p^p(\mu,\nu)\right|^2\right\}^{1/2} \\
&\leq L^{-1/2}\left\{\int_{\mathbb{S}^{d-1}}\left\{\mathbf{\Delta}^p(\theta_\sharp^\star\mu,\theta_\sharp^\star\nu) - \mathbf{S\Delta}_p^p(\mu,\nu)\right\}^2\mathrm{d}\boldsymbol{\sigma}(\theta)\right\}^{1/2} ,
\end{aligned}
$$

Since $\mathbf{S\Delta}_p^p(\mu,\nu) = \int_{\mathbb{S}^{d-1}}\mathbf{\Delta}^p(\theta_\sharp^\star\mu,\theta_\sharp^\star\nu)\mathrm{d}\boldsymbol{\sigma}(\theta)$ by definition, the quantity $\int_{\mathbb{S}^{d-1}}\left\{\mathbf{\Delta}^p(\theta_\sharp^\star\mu,\theta_\sharp^\star\nu) - \mathbf{S\Delta}_p^p(\mu,\nu)\right\}^2\mathrm{d}\boldsymbol{\sigma}(\theta)$ is the variance of $\mathbf{\Delta}^p(\theta_\sharp^\star\mu,\theta_\sharp^\star\nu)$ with respect to $\theta\sim\boldsymbol{\sigma}$.

$\square$

## S1.9 The overall complexity

We now leverage Theorems 4 and 6 to derive the *overall complexity* of sliced divergences, *i.e.* the convergence rate of $\widehat{\mathbf{S\Delta}}_p(\hat\mu_n, \hat\nu_n)$ to $\mathbf{S\Delta}_p(\mu, \nu)$. This result is useful as it helps understanding the behavior of sliced divergences in most practical applications, where $\mathbf{S\Delta}_p(\mu, \nu)$ is approximated using finite sets of samples drawn from $\mu$ and $\nu$ along with Monte Carlo estimates.

**Corollary S4.** *Let $p \in [1, \infty)$ and $\mu, \nu \in \mathcal{P}(\mathbb{R}^d)$. Denote by $\hat\mu_n$ (respectively, $\hat\nu_n$) the empirical distribution computed over a sequence of i.i.d. random variables $X_{1:n} = \{X_k\}_{k=1}^n$ from $\mu$ (resp., $Y_{1:n} = \{Y_k\}_{k=1}^n$ from $\nu$). Assume $\mathbf{\Delta}^p$ admits the following sample complexity: for any $\mu', \nu' \in \mathcal{P}(\mathbb{R})$ and empirical instantiations $\hat\mu'_n, \hat\nu'_n$, $\mathbb{E}[|\mathbf{\Delta}^p(\mu', \nu') - \mathbf{\Delta}^p(\hat\mu'_n, \hat\nu'_n)|] \leq \beta(p, n)$. Then,*

$$\mathbb{E}\big[|\widehat{\mathbf{S\Delta}}_{p,L}^p(\hat\mu_n, \hat\nu_n) - \mathbf{S\Delta}_p^p(\mu, \nu)|\big] \leq \beta(p, n)$$

$$+ L^{-1/2} \left[ \int_{\mathbb{S}^{d-1}} \mathbb{E}\left[ \big(\mathbf{\Delta}^p(\theta_\sharp^\star \hat\mu_n, \theta_\sharp^\star \hat\nu_n) - \mathbf{S\Delta}_p^p(\hat\mu_n, \hat\nu_n)\big)^2 \right] \mathrm{d}\boldsymbol{\sigma}(\theta) \right]^{1/2},$$

*where $\widehat{\mathbf{S\Delta}}_{p,L}^p(\hat\mu_n, \hat\nu_n)$ is defined by (5), and $\mathbb{E}$ is the expectation with respect to (w.r.t.) $X_{1:n}$, $Y_{1:n}$ and $\{\theta_l\}_{l=1}^L$ i.i.d. from the uniform distribution on $\mathbb{S}^{d-1}$.*

*Proof of Corollary S4.* Let $p \in [1, \infty)$, $\mu, \nu \in \mathcal{P}(\mathbb{R}^d)$ and the respective empirical distributions $\hat\mu_n, \hat\nu_n$. By the triangle inequality,

$$|\widehat{\mathbf{S\Delta}}_{p,L}^p(\hat\mu_n, \hat\nu_n) - \mathbf{S\Delta}_p^p(\mu, \nu)| \leq |\widehat{\mathbf{S\Delta}}_{p,L}^p(\hat\mu_n, \hat\nu_n) - \mathbf{S\Delta}_p^p(\hat\mu_n, \hat\nu_n)| + |\mathbf{S\Delta}_p^p(\hat\mu_n, \hat\nu_n) - \mathbf{S\Delta}_p^p(\mu, \nu)|.$$

Therefore, by linearity of expectation, we have

$$\mathbb{E}\big[|\widehat{\mathbf{S\Delta}}_{p,L}^p(\hat\mu_n, \hat\nu_n) - \mathbf{S\Delta}_p^p(\mu, \nu)|\big]$$
$$\leq \mathbb{E}\left[ \mathbb{E}[|\widehat{\mathbf{S\Delta}}_{p,L}^p(\hat\mu_n, \hat\nu_n) - \mathbf{S\Delta}_p^p(\hat\mu_n, \hat\nu_n)| \big| X_{1:n}, Y_{1:n}] \right] + \mathbb{E}\left[ |\mathbf{S\Delta}_p^p(\hat\mu_n, \hat\nu_n) - \mathbf{S\Delta}_p^p(\mu, \nu)| \right].$$
$$\text{(S33)}$$

We bound the left term in (S33). By Theorem 6, we have

$$\mathbb{E}\big[|\widehat{\mathbf{S\Delta}}_{p,L}^p(\hat\mu_n, \hat\nu_n) - \mathbf{S\Delta}_p^p(\hat\mu_n, \hat\nu_n)| \,\big|\, X_{1:n}, Y_{1:n}\big]$$
$$\leq L^{-1/2} \left\{ \int_{\mathbb{S}^{d-1}} \big\{ \mathbf{\Delta}^p(\theta_\sharp^\star \hat\mu_n, \theta_\sharp^\star \hat\nu_n) - \mathbf{S\Delta}_p^p(\hat\mu_n, \hat\nu_n) \big\}^2 \mathrm{d}\boldsymbol{\sigma}(\theta) \right\}^{1/2}.$$

By taking the expectation then using Jensen's inequality, we get

$$\mathbb{E}\left[ \mathbb{E}\big[|\widehat{\mathbf{S\Delta}}_{p,L}^p(\hat\mu_n, \hat\nu_n) - \mathbf{S\Delta}_p^p(\hat\mu_n, \hat\nu_n)| \,\big|\, X_{1:n}, Y_{1:n}\big] \right]$$
$$\leq L^{-1/2} \mathbb{E}\left[ \left\{ \int_{\mathbb{S}^{d-1}} \big\{ \mathbf{\Delta}^p(\theta_\sharp^\star \hat\mu_n, \theta_\sharp^\star \hat\nu_n) - \mathbf{S\Delta}_p^p(\hat\mu_n, \hat\nu_n) \big\}^2 \mathrm{d}\boldsymbol{\sigma}(\theta) \right\}^{1/2} \right]$$
$$\leq L^{-1/2} \mathbb{E}^{1/2}\left[ \int_{\mathbb{S}^{d-1}} \big\{ \mathbf{\Delta}^p(\theta_\sharp^\star \hat\mu_n, \theta_\sharp^\star \hat\nu_n) - \mathbf{S\Delta}_p^p(\hat\mu_n, \hat\nu_n) \big\}^2 \mathrm{d}\boldsymbol{\sigma}(\theta) \right]. \quad \text{(S34)}$$

Next, we bound the right term in (S33): by the sample complexity assumption for $\mathbf{\Delta}^p$ and Theorem 4, we have

$$\mathbb{E}\left[ |\mathbf{S\Delta}_p^p(\hat\mu_n, \hat\nu_n) - \mathbf{S\Delta}_p^p(\mu, \nu)| \right] \leq \beta(p, n). \quad \text{(S35)}$$

Combining (S34) and (S35) in (S33) completes the proof.

$$\square$$

**Remark 1.** *Note that by Fubini's theorem, $\int_{\mathbb{S}^{d-1}} \mathbb{E}[(\mathbf{\Delta}^p(\theta_\sharp^\star \hat\mu_n, \theta_\sharp^\star \hat\nu_n) - \mathbf{S\Delta}_p^p(\hat\mu_n, \hat\nu_n))^2] \mathrm{d}\boldsymbol{\sigma}(\theta)$ (which appears in Corollary S4) is equal to $\mathbb{E}[\mathrm{Var}\{\mathbf{\Delta}^p(\theta_\sharp^\star \hat\mu_n, \theta_\sharp^\star \hat\nu_n)|X_{1:n}, Y_{1:n}\}]$, where $\mathrm{Var}$ is the variance w.r.t. $X_{1:n}$, $Y_{1:n}$ and $\theta$ (which is distributed according to the uniform distribution on $\mathbb{S}^{d-1}$ and independent of $X_{1:n}, Y_{1:n}$).*

## S2 Postponed proofs for Section 4

### S2.1 Applications of Theorem 1

As discussed in Section 4, we can use the general result in Theorem 1 to establish novel topological properties for specific sliced probability divergences, for example the Sliced-Cramér distance (whose definition is recalled in Definition S2) and the broader class of Sliced-IPMs. We present our results and proofs for these examples below.

**Definition S1** (Cramér distance [8]). *Let* $p \in [1, \infty)$ *and* $\mu, \nu \in \mathcal{P}(\mathbb{R})$. *Denote by* $F_\mu, F_\nu$ *the cumulative distribution functions of* $\mu, \nu$ *respectively. The Cramér distance of order* $p$ *between* $\mu$ *and* $\nu$ *is defined by*

$$\mathbf{C}_p^p(\mu, \nu) = \int_{\mathbb{R}} |F_\mu(t) - F_\nu(t)|^p \, \mathrm{d}t \ .$$

**Definition S2** (Sliced-Cramér distance [9]). *Let* $p \in [1, \infty)$ *and* $\mu, \nu \in \mathcal{P}(\mathbb{R}^d)$. *The Sliced-Cramér distance of order* $p$ *between* $\mu$ *and* $\nu$ *is defined by*

$$\mathbf{SC}_p^p(\mu, \nu) = \int_{\mathbb{S}^{d-1}} \mathbf{C}_p^p(\theta_\sharp^\star \mu, \theta_\sharp^\star \nu) \mathrm{d}\boldsymbol{\sigma}(\theta) \ .$$

**Corollary S5.** *Let* $p \in [1, \infty)$. *For any sequence* $(\mu_k)_{k \in \mathbb{N}}$ *in* $\mathcal{P}(\mathbb{R}^d)$ *and* $\mu \in \mathcal{P}(\mathbb{R}^d)$, $\lim_{k \to \infty} \mathbf{SC}_p(\mu_k, \mu) = 0$ *implies* $(\mu_k)_{k \in \mathbb{N}}$ *converges weakly to* $\mu$. *Besides, if* $(\mu_k)_{k \in \mathbb{N}}$ *and* $\mu$ *are supported on a compact space* $\mathsf{K} \subset \mathbb{R}^d$, *then the converse implication holds, meaning that the convergence under* $\mathbf{SC}_p$ *is equivalent to the weak convergence in* $\mathcal{P}(\mathsf{K})$.

*Proof.* Let $p \in [1, \infty)$. By Hölder's inequality, for any $\mu', \nu' \in \mathcal{P}(\mathbb{R})$, we have

$$\mathbf{C}_1(\mu', \nu') \leq \mathbf{C}_p(\mu', \nu') \ . \tag{S36}$$

Consider a sequence $(\mu'_k)_{k \in \mathbb{N}}$ in $\mathcal{P}(\mathbb{R})$ and $\mu' \in \mathcal{P}(\mathbb{R})$ such that $\lim_{k \to \infty} \mathbf{C}_p(\mu'_k, \mu') = 0$. By (S36), this implies $\lim_{k \to \infty} \mathbf{C}_1(\mu'_k, \mu') = 0$. Since the Cramér distance of order 1 is equivalent to the Wasserstein distance of order 1, then by [10, Theorem 6.8], the convergence of $(\mu'_k)_{k \in \mathbb{N}}$ to $\mu'$ under $\mathbf{C}_p$ implies $(\mu'_k)_{k \in \mathbb{N}}$ converges weakly to $\mu'$ in $\mathcal{P}(\mathbb{R})$. By Theorem 1, we conclude that the convergence under $\mathbf{SC}_p$ implies the weak convergence in $\mathcal{P}(\mathbb{R}^d)$.

We now show the second part of the statement. This result partly follows from slight modifications of the techniques we used in the proof of Theorem 1. Consider a compact space $\mathsf{K}' \subset \mathbb{R}$ and a sequence $(\mu'_k)_{k \in \mathbb{N}}$ in $\mathcal{P}(\mathsf{K}')$. Suppose that $(\mu'_k)_{k \in \mathbb{N}}$ converges weakly to $\mu' \in \mathcal{P}(\mathsf{K}')$. Since $F_{\mu'}$ is non-decreasing, it is almost everywhere continuous w.r.t. to the Lebesgue convergence, and using the Portmanteau theorem, we get that for Leb-almost every $t \in \mathbb{R}$, $\lim_{k \to \infty} F_{\mu'_k}(t) = F_{\mu'}(t)$. Besides, for any $k \in \mathbb{N}$ and $t \in \mathsf{K}'$, $|F_{\mu'_k}(t)| \leq 1$, and since $\mathsf{K}'$ is compact, $\left(\int_{\mathsf{K}'} 1^p \mathrm{d}t\right)^{1/p} < \infty$. By the dominated convergence theorem in $\mathrm{L}^p$-spaces, we conclude that

$$\lim_{k \to \infty} \left\{ \int_{\mathsf{K}'} |F_{\mu'_k}(t) - F_{\mu'}(t)|^p \mathrm{d}t \right\}^{1/p} = 0 \ , \tag{S37}$$

in other words, the weak convergence of measures in $\mathcal{P}(\mathsf{K}')$, where $\mathsf{K}'$ is a compact subspace of $\mathbb{R}$, implies the convergence under $\mathbf{C}_p$.

Now, consider a compact space $\mathsf{K} \subset \mathbb{R}^d$ and a sequence $(\mu_k)_{k \in \mathbb{N}}$ in $\mathcal{P}(\mathsf{K})$ which converges weakly to $\mu \in \mathcal{P}(\mathsf{K})$. For any $\theta \in \mathbb{S}^{d-1}$, define $\mathsf{K}_\theta = \{\langle \theta, x \rangle : x \in \mathsf{K}\}$, which is a compact subset of $\mathbb{R}$ (since it is the image of $\mathsf{K}$ by a continuous function) with $\mathrm{diam}(\mathsf{K}_\theta) \leq \mathrm{diam}(\mathsf{K})$ (by the Cauchy-Schwarz inequality). The sequence of pushforward measures $(\theta_\sharp^\star \mu_k)_{k \in \mathbb{N}}$ is in $\mathcal{P}(\mathsf{K}_\theta)$ and, by the continuous mapping theorem, converges weakly to $\theta_\sharp^\star \mu \in \mathcal{P}(\mathsf{K}_\theta)$. Therefore, by (S37), for any $\theta \in \mathbb{S}^{d-1}$,

$$\lim_{k \to \infty} \mathbf{C}_p(\theta_\sharp^\star \mu_k, \theta_\sharp^\star \mu) = 0 \ . \tag{S38}$$

Besides, for any $\mu, \nu \in \mathcal{P}(\mathbb{R}^d)$ with support in $\mathsf{K}$, and $\theta \in \mathbb{S}^{d-1}$,

$$\mathbf{C}_p(\theta_\sharp^\star \nu, \theta_\sharp^\star \mu) = \int_{\mathbb{R}} |F_\nu(t) - F_\mu(t)|^p \, \mathrm{d}t = \int_{\mathsf{K}_\theta} |F_\nu(t) - F_\mu(t)|^p \, \mathrm{d}t$$

$$\leq 2^p \mathrm{diam}(\mathsf{K}_\theta) \leq 2^p \mathrm{diam}(\mathsf{K}) \ .$$

By (S38) and the dominated convergence theorem, we finally obtain $\lim_{k\to\infty} \mathbf{SC}_p(\mu_k, \mu) = 0$.

$\square$

**Corollary S6.** *Let $p \in [1, \infty)$ and $\widetilde{\mathsf{F}} \subset \mathbb{M}_b(\mathbb{R})$. Suppose that the space spanned by $\widetilde{\mathsf{F}}$ is dense in the space of continous functions for $\|\cdot\|_\infty$. Then, the convergence under the Sliced Integral Probability Metric of order $p$ associated with $\widetilde{\mathsf{F}}$, $\mathbf{S}\gamma_{\widetilde{\mathsf{F}},p}$, implies the weak convergence in $\mathcal{P}(\mathbb{R}^d)$. Besides, if $\gamma_{\widetilde{\mathsf{F}}}$ is bounded, the converge implication holds, i.e. the weak convergence in $\mathcal{P}(\mathbb{R}^d)$ implies the convergence under $\mathbf{S}\gamma_{\widetilde{\mathsf{F}},p}$.*

*Proof.* By construction of $\widetilde{\mathsf{F}}$ and [11, Section 5.1], $\gamma_{\widetilde{\mathsf{F}}}$ metrizes the weak convergence in $\mathcal{P}(\mathbb{R})$, *i.e.* the weak convergence in $\mathcal{P}(\mathbb{R})$ is equivalent to the convergence of measures under $\gamma_{\widetilde{\mathsf{F}}}$. The properties of $\mathbf{S}\gamma_{\widetilde{\mathsf{F}},p}$, $p \in [1, \infty)$ result from the application of Theorem 1.

$\square$

**Remark 2.** *The boundedness assumption for $\gamma_{\widetilde{\mathsf{F}}}$ is achieved if we additionally suppose that $\widetilde{\mathsf{F}}$ is a uniformly bounded family of functions in $\mathbb{M}(\mathbb{R})$, which is a mild assumption.*

## S2.2 Proof of Corollary 2

**Lemma S2.** *Let $p \in [1, \infty)$ and $\mu' \in \mathcal{P}(\mathbb{R})$ with empirical distribution $\hat{\mu}'_n$. Suppose there exists $q > p$ such that the moment of order $q$ of $\mu'$, defined as $M_q(\mu') = \int_{\mathbb{R}} |t|^q \, \mathrm{d}\mu'(t)$, is bounded above by $K < \infty$. Then, there exists a constant $C_{p,q}$ depending on $p, q$ such that*

$$\mathbb{E}\left[\mathbf{W}_p^p(\hat{\mu}'_n, \mu')\right] \leq C_{p,q} K \begin{cases} n^{-1/2} & \text{if } q > 2p, \\ n^{-1/2} \log(n) & \text{if } q = 2p, \\ n^{-(q-p)/q} & \text{if } q \in (p, 2p). \end{cases}$$

*Proof.* This immediately results from [12, Theorem 1]. $\square$

*Proof of Corollary 2.* We first recall that, for any $\xi \in \mathcal{P}(\mathbb{R}^s)$ ($s \geq 1$) and $\theta \in \mathbb{S}^{d-1}$, the moment of order $k > 0$ of $\theta^\star_\sharp \xi$ is lower than the one associated with $\xi$. Indeed, by using the property of pushforward measures, the Cauchy-Schwarz inequality, and $\|\theta\| \leq 1$, we have

$$M_k(\theta^\star_\sharp \xi) = \int_{\mathbb{R}} |t|^k \, \mathrm{d}\theta^\star_\sharp \xi(t) = \int_{\mathbb{R}^d} |\langle \theta, x \rangle|^k \, \mathrm{d}\xi(x) \leq \int_{\mathbb{R}^d} \|x\|^k \, \mathrm{d}\xi(x) = M_k(\xi) . \tag{S39}$$

Now, let $p \in [1, \infty)$ and $\mu \in \mathcal{P}_q(\mathbb{R}^d)$ ($q > p$) with empirical distribution $\hat{\mu}_n$. Then, by (S39), for any $\theta \in \mathbb{S}^{d-1}$, $M_q(\theta^\star_\sharp \mu) \leq M_q(\mu) < \infty$, and we can apply Lemma S2 and Theorem 5 to derive the convergence rate under $\mathbf{SW}_p$ : there exists a constant $C_{p,q}$ such that,

$$\mathbb{E}\left[\mathbf{SW}_p^p(\hat{\mu}_n, \mu)\right] \leq C_{p,q} M_q^{p/q}(\mu) \begin{cases} n^{-1/2} & \text{if } q > 2p, \\ n^{-1/2} \log(n) & \text{if } q = 2p, \\ n^{-(q-p)/q} & \text{if } q \in (p, 2p). \end{cases} \tag{S40}$$

Besides, since $\mathbf{W}_p$ is a metric, we can apply Theorem 5 to derive the sample complexity of $\mathbf{SW}_p$. Consider $\mu, \nu \in \mathcal{P}_q(\mathbb{R}^d)$ with $q > p$, with respective empirical measures $\hat{\mu}_n, \hat{\nu}_n$. Then, starting from (S31) and using the convergence rate derived in (S40), we obtain the desired result as follows

$$\begin{aligned} &\mathbb{E}\left|\mathbf{SW}_p(\mu, \nu) - \mathbf{SW}_p(\hat{\mu}_n, \hat{\nu}_n)\right| \\ &\leq \left\{\mathbb{E}\left|\mathbf{SW}_p^p(\hat{\mu}_n, \mu)\right|\right\}^{1/p} + \left\{\mathbb{E}\left|\mathbf{SW}_p^p(\hat{\nu}_n, \nu)\right|\right\}^{1/p} \\ &\leq C_{p,q}^{1/p} \left(M_q^{1/q}(\mu) + M_q^{1/q}(\nu)\right) \begin{cases} n^{-1/(2p)} & \text{if } q > 2p, \\ n^{-1/(2p)} \log(n)^{1/p} & \text{if } q = 2p, \\ n^{-(q-p)/(pq)} & \text{if } q \in (p, 2p). \end{cases} \end{aligned}$$

$\square$

## S2.3 Proof of Theorem 7

*Proof of Theorem 7.* Let $p \in [1, \infty)$ and $\varepsilon \geq 0$. We use the reformulation of $\mathbf{W}_{p,\varepsilon}$ as the maximum of an expectation, as given in [13, Proposition 2.1],

$$
\begin{aligned}
\mathbf{SW}_{p,\varepsilon}^p(\mu, \nu) &= \int_{\mathbb{S}^{d-1}} \mathbf{W}_{p,\varepsilon}^p(\theta_\sharp^\star \mu, \theta_\sharp^\star \nu) \mathrm{d}\boldsymbol{\sigma}(\theta) \\
&= \int_{\mathbb{S}^{d-1}} \left\{ \max_{\tilde{u}, \tilde{v} \in \mathrm{C}(\mathbb{R})} \mathbb{E}_{\theta_\sharp^\star \mu \otimes \theta_\sharp^\star \nu} \left[ \phi_\varepsilon \left( \tilde{u}(\tilde{X}), \tilde{v}(\tilde{Y}), \tilde{X}, \tilde{Y} \right) \right] \right\}^p \mathrm{d}\boldsymbol{\sigma}(\theta) ,
\end{aligned}
\tag{S41}
$$

where $\mathrm{C}(\mathbb{R})$ denotes the set of continuous real functions, and $\phi_\varepsilon(t, s, x, y) = t + s - \varepsilon e^{(t+s-\|x-y\|^p)/\varepsilon}$.

Consider for any $\theta \in \mathbb{S}^{d-1}$, $\tilde{u}_\theta^\star, \tilde{v}_\theta^\star$ as the functions attaining the maximum in (S41), which exist by [14, Theorem 4 in the supplementary document]. We obtain

$$
\begin{aligned}
\mathbf{SW}_{p,\varepsilon}^p(\mu, \nu) &= \int_{\mathbb{S}^{d-1}} \left\{ \mathbb{E}_{\theta_\sharp^\star \mu \otimes \theta_\sharp^\star \nu} \left[ \phi_\varepsilon \left( \tilde{u}_\theta^\star(\tilde{X}), \tilde{v}_\theta^\star(\tilde{Y}), \tilde{X}, \tilde{Y} \right) \right] \right\}^p \mathrm{d}\boldsymbol{\sigma}(\theta) \\
&= \int_{\mathbb{S}^{d-1}} \left\{ \mathbb{E}_{\mu \otimes \nu} \left[ \phi_\varepsilon \left( \tilde{u}_\theta^\star \circ \theta^\star(X), \tilde{v}_\theta^\star \circ \theta^\star(Y), X, Y \right) \right] \right\}^p \mathrm{d}\boldsymbol{\sigma}(\theta) .
\end{aligned}
\tag{S42}
$$

Since for all $\tilde{w} \in \mathrm{C}(\mathbb{R})$ and $\theta \in \mathbb{S}^{d-1}$, $\tilde{w} \circ \theta^\star \in \mathrm{C}(\mathbb{R}^d)$, we can bound (S42) as follows

$$
\mathbf{SW}_{p,\varepsilon}^p(\mu, \nu) \leq \int_{\mathbb{S}^{d-1}} \left\{ \max_{u, v \in \mathrm{C}(\mathbb{R}^d)} \mathbb{E}_{\mu \otimes \nu} \left[ \phi_\varepsilon \left( u(X), v(Y), X, Y \right) \right] \right\}^p \mathrm{d}\boldsymbol{\sigma}(\theta) = \mathbf{W}_{p,\varepsilon}^p(\mu, \nu) .
\tag{S43}
$$

By Proposition 1, since $\mathbf{W}_{p,\varepsilon}$ is non-negative, so is $\mathbf{SW}_{p,\varepsilon}$, and we can apply $t \mapsto t^{1/p}$ on both sides of (S43) to obtain the final result.

$\square$

## S2.4 Proof of Theorem 8

**Proposition S1.** *Let $\tilde{\mathsf{X}}$ be a compact subset of $\mathbb{R}$, and $\mu', \nu' \in \mathcal{P}(\tilde{\mathsf{X}})$ with respective empirical instantiations $\hat{\mu}_n', \hat{\nu}_n'$. Let $p \in [1, \infty)$ and $\varepsilon \geq 0$. Then,*

$$
|\mathbf{W}_{p,\varepsilon}(\hat{\mu}_n', \hat{\nu}_n') - \mathbf{W}_{p,\varepsilon}(\mu', \nu')| \leq 2 \operatorname{diam}(\tilde{\mathsf{X}}) \left\{ \mathbf{W}_1(\mu', \hat{\mu}_n') + \mathbf{W}_1(\nu', \hat{\nu}_n') \right\} .
\tag{S44}
$$

*Proof.* Let $p \in [1, \infty)$, $\varepsilon \geq 0$ and $\tilde{\mathsf{X}} \subset \mathbb{R}$ compact. Consider $\mu', \nu' \in \mathcal{P}(\tilde{\mathsf{X}})$ with respective empirical distributions $\hat{\mu}_n', \hat{\nu}_n'$. We first express the regularized OT cost as the maximum of an expectation [13, Proposition 2.1]

$$
\mathbf{W}_{p,\varepsilon}(\mu', \nu') = \max_{\tilde{u}, \tilde{v} \in \mathrm{C}(\mathbb{R})} \mathbb{E}_{\mu' \otimes \nu'} \left[ \phi_\varepsilon \left( \tilde{u}(\tilde{X}), \tilde{v}(\tilde{Y}), \tilde{X}, \tilde{Y} \right) \right]
\tag{S45}
$$

$$
\mathbf{W}_{p,\varepsilon}(\hat{\mu}_n', \nu') = \max_{\tilde{u}, \tilde{v} \in \mathrm{C}(\mathbb{R})} \mathbb{E}_{\hat{\mu}_n' \otimes \nu'} \left[ \phi_\varepsilon \left( \tilde{u}(\tilde{X}), \tilde{v}(\tilde{Y}), \tilde{X}, \tilde{Y} \right) \right] ,
\tag{S46}
$$

where $\phi_\varepsilon(t, s, x, y) = t + s - \varepsilon e^{(t+s-\|x-y\|^2/2)/\varepsilon}$. By [14, Proposition 1], the Sinkhorn potentials $(\tilde{u}, \tilde{v})$ are Lipschitz continuous with Lipschitz constant $\operatorname{diam}(\tilde{\mathsf{X}}) < \infty$. Therefore, by denoting by $\mathrm{Lip}_{\operatorname{diam}(\tilde{\mathsf{X}})}(\mathbb{R})$ the space of $\operatorname{diam}(\tilde{\mathsf{X}})$-Lipschitz continuous functions defined on $\mathbb{R}$, (S45) and (S46) can be rewritten with the maximization over $\mathrm{Lip}_{\operatorname{diam}(\tilde{\mathsf{X}})}(\mathbb{R})$.

We can now use [15, Proposition 2] to bound the absolute difference of $\mathbf{W}_{p,\varepsilon}(\mu', \nu')$ and $\mathbf{W}_{p,\varepsilon}(\hat{\mu}_n', \nu')$. We provide the detailed proof below for completeness. By [15, Proposition 6, Appendix A], there exist smooth potentials $(\tilde{u}^\star, \tilde{v}^\star)$ attaining the maximum in (S45) such that, for all

$\tilde{x}, \tilde{y} \in \mathbb{R}$,

$$\int_{\mathbb{R}} \phi_\varepsilon(\tilde{u}^\star(\tilde{x}), \tilde{v}^\star(\tilde{y}), \tilde{x}, \tilde{y}) \mathrm{d}\nu'(\tilde{y}) = 1 \quad \mu'\text{-almost surely,} \tag{S47}$$

$$\int_{\mathbb{R}} \phi_\varepsilon(\tilde{u}^\star(\tilde{x}), \tilde{v}^\star(\tilde{y}), \tilde{x}, \tilde{y}) \mathrm{d}\mu'(\tilde{x}) = 1 \quad \nu'\text{-almost surely .} \tag{S48}$$

Analogously, there exist smooth optimal potentials $(\tilde{u}_n^\star, \tilde{v}_n^\star)$ for (S46) satisfying (S47) and (S48) where $\tilde{u}^\star$, $\tilde{v}^\star$ and $\mu'$ are replaced by $\tilde{u}_n^\star$, $\tilde{v}_n^\star$ and $\hat{\mu}_n'$ respectively.

The optimality of these potentials give us

$$\mathbb{E}_{\mu' \otimes \nu'}\left[\phi_\varepsilon(\tilde{u}_n^\star(\tilde{X}), \tilde{v}_n^\star(\tilde{Y}), \tilde{X}, \tilde{Y})\right] - \mathbb{E}_{\hat{\mu}_n' \otimes \nu'}\left[\phi_\varepsilon(\tilde{u}_n^\star(\tilde{X}), \tilde{v}_n^\star(\tilde{Y}), \tilde{X}, \tilde{Y})\right]$$
$$\leq \mathbb{E}_{\mu' \otimes \nu'}\left[\phi_\varepsilon(\tilde{u}^\star(\tilde{X}), \tilde{v}^\star(\tilde{Y}), \tilde{X}, \tilde{Y})\right] - \mathbb{E}_{\hat{\mu}_n' \otimes \nu'}\left[\phi_\varepsilon(\tilde{u}_n^\star(\tilde{X}), \tilde{v}_n^\star(\tilde{Y}), \tilde{X}, \tilde{Y})\right]$$
$$\leq \mathbb{E}_{\mu' \otimes \nu'}\left[\phi_\varepsilon(\tilde{u}^\star(\tilde{X}), \tilde{v}^\star(\tilde{Y}), \tilde{X}, \tilde{Y})\right] - \mathbb{E}_{\hat{\mu}_n' \otimes \nu'}\left[\phi_\varepsilon(\tilde{u}^\star(\tilde{X}), \tilde{v}^\star(\tilde{Y}), \tilde{X}, \tilde{Y})\right] .$$

Therefore,

$$|\mathbf{W}_{p,\varepsilon}(\mu', \nu') - \mathbf{W}_{p,\varepsilon}(\hat{\mu}_n', \nu')|$$
$$= \left|\mathbb{E}_{\mu' \otimes \nu'}\left[\phi_\varepsilon(\tilde{u}^\star(\tilde{X}), \tilde{v}^\star(\tilde{Y}), \tilde{X}, \tilde{Y})\right] - \mathbb{E}_{\hat{\mu}_n' \otimes \nu'}\left[\phi_\varepsilon(\tilde{u}_n^\star(\tilde{X}), \tilde{v}_n^\star(\tilde{Y}), \tilde{X}, \tilde{Y})\right]\right|$$
$$\leq \left|\mathbb{E}_{\mu' \otimes \nu'}\left[\phi_\varepsilon(\tilde{u}^\star(\tilde{X}), \tilde{v}^\star(\tilde{Y}), \tilde{X}, \tilde{Y})\right] - \mathbb{E}_{\hat{\mu}_n' \otimes \nu'}\left[\phi_\varepsilon(\tilde{u}^\star(\tilde{X}), \tilde{v}^\star(\tilde{Y}), \tilde{X}, \tilde{Y})\right]\right|$$
$$+ \left|\mathbb{E}_{\mu' \otimes \nu'}\left[\phi_\varepsilon(\tilde{u}_n^\star(\tilde{X}), \tilde{v}_n^\star(\tilde{Y}), \tilde{X}, \tilde{Y})\right] - \mathbb{E}_{\hat{\mu}_n' \otimes \nu'}\left[\phi_\varepsilon(\tilde{u}_n^\star(\tilde{X}), \tilde{v}_n^\star(\tilde{Y}), \tilde{X}, \tilde{Y})\right]\right| . \tag{S49}$$

We bound each term of the sum in (S49) as follows

$$\left|\mathbb{E}_{\mu' \otimes \nu'}\left[\phi_\varepsilon(\tilde{u}^\star(\tilde{X}), \tilde{v}^\star(\tilde{Y}), \tilde{X}, \tilde{Y})\right] - \mathbb{E}_{\hat{\mu}_n' \otimes \nu'}\left[\phi_\varepsilon(\tilde{u}^\star(\tilde{X}), \tilde{v}^\star(\tilde{Y}), \tilde{X}, \tilde{Y})\right]\right|$$
$$= \left|\int_{\mathbb{R}} \tilde{u}^\star(\tilde{x}) \mathrm{d}(\mu' - \hat{\mu}_n')(\tilde{x}) - \varepsilon \int_{\mathbb{R}} \int_{\mathbb{R}} e^{(\tilde{u}^\star(\tilde{x}) + \tilde{v}^\star(\tilde{y}) - |\tilde{x} - \tilde{y}|^2/2)/\varepsilon} \mathrm{d}\nu'(\tilde{y}) \mathrm{d}(\mu' - \hat{\mu}_n')(\tilde{x})\right|$$
$$= \left|\int_{\mathbb{R}} \tilde{u}^\star(\tilde{x}) \mathrm{d}(\mu' - \hat{\mu}_n')(\tilde{x})\right| \leq \sup_{\tilde{u} \in \mathrm{Lip}_{\mathrm{diam}(\tilde{X})}(\mathbb{R})} \left|\int_{\mathbb{R}} \tilde{u}(\tilde{x}) \mathrm{d}(\mu' - \hat{\mu}_n')(\tilde{x})\right| , \tag{S50}$$

where (S50) results from (S47). Since for any $f \in \mathrm{Lip}_{\mathrm{L}}(\mathbb{R})$ with $\mathrm{L} > 0$, $f/\mathrm{L} \in \mathrm{Lip}_1(\mathbb{R})$, (S50) can be bounded as follows

$$\left|\mathbb{E}_{\mu' \otimes \nu'}\left[\phi_\varepsilon(\tilde{u}^\star(\tilde{X}), \tilde{v}^\star(\tilde{Y}), \tilde{X}, \tilde{Y})\right] - \mathbb{E}_{\hat{\mu}_n' \otimes \nu'}\left[\phi_\varepsilon(\tilde{u}^\star(\tilde{X}), \tilde{v}^\star(\tilde{Y}), \tilde{X}, \tilde{Y})\right]\right|$$
$$\leq \mathrm{diam}(\tilde{X}) \sup_{\tilde{u} \in \mathrm{Lip}_1(\mathbb{R})} \left|\int_{\mathbb{R}} \tilde{u}(\tilde{x}) \mathrm{d}(\theta_\sharp^\star \mu - \theta_\sharp^\star \hat{\mu}_n)(\tilde{x})\right| = \mathrm{diam}(\tilde{X}) \mathbf{W}_1(\mu', \hat{\mu}_n') , \tag{S51}$$

where (S51) follows from the dual formulation of the Wasserstein distance of order 1 [10, Theorem 5.10].

We show with an analogous proof that

$$\left|\mathbb{E}_{\mu' \otimes \nu'}\left[\phi_\varepsilon(\tilde{u}_n^\star(\tilde{X}), \tilde{v}_n^\star(\tilde{Y}), \tilde{X}, \tilde{Y})\right] - \mathbb{E}_{\hat{\mu}_n' \otimes \nu'}\left[\phi_\varepsilon(\tilde{u}_n^\star(\tilde{X}), \tilde{v}_n^\star(\tilde{Y}), \tilde{X}, \tilde{Y})\right]\right| \leq \mathrm{diam}(\tilde{X}) \mathbf{W}_1(\mu', \hat{\mu}_n') ,$$

which leads to the conclusion that

$$|\mathbf{W}_{p,\varepsilon}(\mu', \nu') - \mathbf{W}_{p,\varepsilon}(\hat{\mu}_n', \nu')| \leq 2 \, \mathrm{diam}(\tilde{X}) \mathbf{W}_1(\mu', \hat{\mu}_n') . \tag{S52}$$

By using the triangle inequality and (S52), we obtain the final result

$$|\mathbf{W}_{p,\varepsilon}(\hat{\mu}_n', \hat{\nu}_n') - \mathbf{W}_{p,\varepsilon}(\mu', \nu')| \leq |\mathbf{W}_{p,\varepsilon}(\mu', \nu') - \mathbf{W}_{p,\varepsilon}(\hat{\mu}_n', \nu')| + |\mathbf{W}_{p,\varepsilon}(\hat{\mu}_n', \nu') - \mathbf{W}_{p,\varepsilon}(\hat{\mu}_n', \hat{\nu}_n')|$$
$$\leq 2 \, \mathrm{diam}(\tilde{X}) \{\mathbf{W}_1(\mu', \hat{\mu}_n') + \mathbf{W}_1(\nu', \hat{\nu}_n')\} .$$

$\square$

**Corollary S7.** *Let* $\tilde{X}$ *be a compact subset of* $\mathbb{R}$, *and* $\mu', \nu' \in \mathcal{P}(\tilde{X})$. *Denote by* $\hat{\mu}'_n, \hat{\nu}'_n$ *their respective empirical instantiations. Let* $p \in [1, \infty)$ *and* $\varepsilon \geq 0$. *Then,*

$$\mathbb{E} \left| \mathbf{W}_{p,\varepsilon}(\hat{\mu}'_n, \hat{\nu}'_n) - \mathbf{W}_{p,\varepsilon}(\mu', \nu') \right| \leq 2 \operatorname{diam}(\tilde{X}) C_q \left[ M_q^{1/q}(\mu') + M_q^{1/q}(\nu') \right] n^{-1/2} ,$$

*where* $q > 2$, $C_q < \infty$ *is a constant that depends on* $q$, *and* $M_q(\mu'), M_q(\nu')$ *are the moments of order* $q$ *of* $\mu', \nu'$ *respectively.*

*Proof.* We apply Proposition S1 and take the expectation of (S44) with respect to $\tilde{X}_{1:n} \sim \hat{\mu}'_n$ and $\tilde{Y}_{1:n} \sim \hat{\nu}'_n$

$$\mathbb{E} \left| \mathbf{W}_{p,\varepsilon}(\hat{\mu}'_n, \hat{\nu}'_n) - \mathbf{W}_{p,\varepsilon}(\mu', \nu') \right| \leq 2 \operatorname{diam}(\tilde{X}) \mathbb{E} \left\{ \mathbf{W}_1(\mu', \hat{\mu}'_n) + \mathbf{W}_1(\nu', \hat{\nu}'_n) \right\} . \tag{S53}$$

Since $\mu'$ and $\nu'$ are both supported on a compact space, they have infinitely many finite moments. We can then bound (S53) using the convergence rate of empirical measures in $\mathbf{W}_1$, recalled in Lemma S2. This concludes the proof.

$\square$

*Proof of Theorem 8.* Let $p \in [1, \infty)$ and $\varepsilon \geq 0$. Consider $\mu, \nu \in \mathcal{P}(\mathsf{X})$ with $\mathsf{X} \subset \mathbb{R}^d$ compact, and denote by $\hat{\mu}_n, \hat{\nu}_n$ their respective empirical distributions.

Let $\theta \in \mathbb{S}^{d-1}$ and define $\mathsf{X}_\theta = \{ \langle \theta, x \rangle : x \in \mathsf{X} \}$. $\mathsf{X}_\theta$ is compact (since $\mathsf{X}$ is compact and $\theta^\star$ is continuous) and verifies $\operatorname{diam}(\mathsf{X}_\theta) \leq \operatorname{diam}(\mathsf{X})$ (by the Cauchy-Schwarz inequality). Besides, by (S39), for any $k > 0$, $M_k(\theta^\star_\sharp \mu) \leq M_k(\mu)$ and $M_k(\theta^\star_\sharp \nu) \leq M_k(\nu)$. By Corollary S7, there exists $C_q < \infty$ which depends on $q > 2$ such that,

$$\mathbb{E} \left| \mathbf{W}_{p,\varepsilon}(\theta^\star_\sharp \hat{\mu}_n, \theta^\star_\sharp \hat{\nu}_n) - \mathbf{W}_{p,\varepsilon}(\theta^\star_\sharp \mu, \theta^\star_\sharp \nu) \right| \leq 2 \operatorname{diam}(\mathsf{X}) C_q \left[ M_q^{1/q}(\mu) + M_q^{1/q}(\nu) \right] n^{-1/2} .$$

The sample complexity of $\mathbf{SW}_{p,\varepsilon}$ is finally obtained by applying Theorem 4.

$\square$

## S2.5 Proof of Proposition 2

Sinkhorn's algorithm refers to an iterative procedure which operates on empirical distributions as follows: consider a cost matrix $C$ between two sets of $n$ samples, and define the matrix $K$ with $K_{i,j} = \exp(-C_{i,j}/\varepsilon)$ for $1 \leq i, j \leq n$, and initialize $b^{(0)} = 1 \in \mathbb{R}^n$ ; then, compute for $\ell > 1$, $a^{(\ell)} = 1./n(Kb^{(\ell-1)}), b^{(\ell)} = 1./n(Ka^{(\ell)})$, where ./ stands for the entry-wise division. This defines a sequence $\gamma_{i,j}^{(\ell)} = a_i^{(\ell)} K_{i,j} b_j^{(\ell)}$, which converges to a solution of (3) at a linear rate. The convergence rate of Sinkhorn's algorithm is recalled in Theorem S1. For an extended discussion on this result, we refer to [16, Section 4.2].

**Theorem S1** ([17]). *The iterates* $a^{(\ell)}$ *and* $b^{(\ell)}$ *of Sinkhorn's algorithm converge linearly for the Hilbert metric at a rate* $1 - \tanh(\tau(K)/4)$, *with* $\tau(K) = \log \max_{i,j,i',j'} \frac{K_{ij} K_{i'j'}}{K_{ij'} K_{i'j}}$. *In particular, for the squared-norm cost, i.e.* $K_{ij} = \exp(-\|x_i - x_j\|^2/\varepsilon)$, *it holds*

$$\tau(K) \leq 2 \max_{i,j} \|x_i - x_j\|^2/\varepsilon.$$

*Proof of Proposition 2.* For $i, j \in \{1, \dots, n\}$, the function $f_{i,j} : \theta \in \mathbb{S}^{d-1} \mapsto \frac{1}{R} \langle \theta, x_i - x_j \rangle$ is 1-Lipschitz and has median 0 for $\theta$ uniformly distributed on the unit sphere. Thus, by concentration of measure on the sphere [18, Example 3.12], it holds for $\varepsilon > 0$,

$$\mathbb{P} \left( |f_{i,j}(\theta)| \geq \varepsilon \right) \leq \sqrt{2\pi} \exp(-d\varepsilon^2/2) .$$

Taking a union bound over the $n(n-1)$ pairs of indices and setting $\tau = (R\varepsilon)^2$, it follows

$$\mathbb{P} \left( \max_{i,j} |\langle \theta, x_i - x_j \rangle|^2 \geq \tau \right) \leq \sqrt{2\pi} n^2 \exp(-d\tau/2R^2) .$$

Hence, for any $\delta > 0$, it holds with probability $1 - \delta$ that $\max_{i,j} |\langle \theta, x_i - x_j \rangle|^2 \leq \frac{2R^2}{d} \log(\sqrt{2\pi} n^2/\delta)$. This argument was suggested to us by an anonymous reviewer.

$\square$

# S3    Additional experimental results

All of our experimental findings presented in this paper and its supplementary document can be reproduced with the code that we provided here: `https://github.com/kimiandj/sliced_div`.

In this section, we provide additional results obtained for the synthetical experiments illustrating the sample complexity of Sliced-Wasserstein and Sliced-Sinkhorn divergences: we produce figures analogously to Figures 2(b), 3(a) and 3(b), with different hyperparameter values.

(a) $L = 1$                    (b) $L = 10$                    (c) $L = 1000$

Figure S1: Illustration of Corollary 2: Wasserstein and Sliced-Wasserstein distances of order 2 between two sets of $n$ samples generated from $\mathcal{N}(\mathbf{0}, \mathbf{I}_d)$ vs. $n$, for different $d$, on log-log scale. $\mathbf{SW}_2$ is approximated with $L$ random projections, $L \in \{1, 10, 1000\}$. Results are averaged over 100 runs, and the shaded areas correspond to the 10th-90th percentiles. Figure 2(b) shows the results for $L = 100$.

(a) Influence of the data dimension for $\varepsilon \in \{0.05, 10, 100\}$

(b) Influence of the regularization coefficient for $d \in \{2, 10, 50\}$

Figure S2: Illustration of Theorem 8: Sinkhorn and Sliced-Sinkhorn divergences between two sets of $n$ samples generated from $\mathcal{N}(\mathbf{0}, \mathbf{I}_d)$ for different values of $n$, dimension $d$, and regularization coefficient $\varepsilon$. Sliced-Sinkhorn is approximated with 10 random projections. Results are averaged over 100 runs, and the shaded areas correspond to the 10th-90th percentiles. All plots have a log-log scale. Figure 3(a) shows the influence of the dimension for $\varepsilon = 1$, and Figure 3(b) shows the influence of the regularization for $d = 100$.

## Footnotes

*Corresponding author: kimia.nadjahi@telecom-paris.fr