[Reviews · NeurIPS 2020]

Review 1

Summary and Contributions: This paper establishes a general theory for sliced probability divergences, which generalizes the popular theory on the Wasserstein distance to a much broader class of divergences including integral probability metrics and Sinkhorn divergences. Topologically, the paper proves that the sliced divergences metrize the weak topology under mild conditions, as well as the topological equivalence between the sliced divergence and the divergence it bases upon when the probability measures have finite supports, Statistically, the paper shows that the convergence rate (aka sample complexity) of the sliced divergences is in the same order as that of the base divergence for one-dimensional measures, thereby avoiding the curse of dimensionality. The appendix provides various extra interesting examples.

Strengths: The paper is well-written and the proofs are clear and well-structured. The unified theory should be of interest in many areas of machine learning and statistics.

Weaknesses: I only have a few minor comments and suggestions. (1) line 168, "We note that the L-Lipschitz assumption is not crucial for this result and can be exchanged with a uniform continuity assumption". Could you elaborate how the Lipschitz assumption can be relaxed? (2) One major advantage of the sliced divergences is the dimension-independent sample complexity. But is it meaningful to compare the sample complexity of a sliced divergence and its based divergence? After all, they are in different scales. (3) line 53, "we prove that the 'sample complexity' of S∆ is proportional to the sample complexity of ∆ and does not depend on the dimension d". This was confusing in my first read. I think it is more precise to say that "the 'sample complexity' of S∆ is proportional to the sample complexity of ∆ for one-dimensional measures...". (4) Proposition 2 can be sharpened using the concentration of measure on the sphere (see e.g. Theorem 1.5 of Vershynin (2011)). In particular, let f_{ij}(\theta) = <x_i - x_j, \theta> / R. Then f_{ij} is 1-Lipschitz with median 0. Since \theta is uniformly distributed on the unite sphere, P(|f_{ij}(\theta)| >= t) <= 2e^{-dt^2 / 2}. Therefore, max_{ij}|f_{ij}(\theta)| <= t^2 with probability 1 - n(n-1)e^{-dt^2 / 2}. This would give a bound O(R^2 / d \log (n / \delta)), instead of the bound O(R^2 / \sqrt{d} \log (n / \delta)). (5) Are there clean examples where S∆ (μ_n ,ν_n ) --> 0, but ∆ (μ_n ,ν_n ) -/-> 0 for TV distance or for other divergences with unbounded domain? Reference: Vershynin, Roman. "Lectures in geometric functional analysis." Unpublished manuscript. Available at http://www-personal. umich. edu/romanv/papers/GFA-book/GFA-book. pdf (2011). ------------------ Post-rebuttal update ------------------- The authors' response clearly addressed all my concerns so I raised my score to 8.

Correctness: Yes

Clarity: Yes

Relation to Prior Work: Yes

Reproducibility: Yes

Additional Feedback:


Review 2

Summary and Contributions: This paper develop and analysis the statistical and topological property of the sliced probability measure based on integral probability metric. It considers the projection of multi-variate probability measure into a particular direction, named Sliced Probability Divergences (SPDs), which can be used to learn a large class of distributions. Theoretical properties are provided.

Strengths: I think this is an interesting paper, addressing the properties of IPM with 1d projection. the paper progressed clearly and both theoretical and empirical results provided are sound.

Weaknesses: For sampling the random projection, uniform distribution used on choosing directions, as stated in Thm 2. This is analogous to say each direction are equally important. When comparing distributions, projections onto some directions are arguably more "important" in terms of distinguishing distributions. will this improve the results? (say thm 2), as in line 198, the complexity between estimated slicing and the population slicing is stated while not explicitly analyzed.

Correctness: I don't find a major flaw in the theory and the empirical examples are correct.

Clarity: I think the paper is well presented.

Relation to Prior Work: The paper had an adequate survey on previous work. As the paper discussed implicit generative model; maybe https://arxiv.org/pdf/1610.03483.pdf and https://arxiv.org/abs/1611.04488 are useful citations. As the paper discussed MMD Wasserstein and Sinkhorn divergence in applications, it may be nice to draw the connections via: https://arxiv.org/abs/1810.02733

Reproducibility: Yes

Additional Feedback: --- after response --- After author's response, I have gain a better understanding about the current stage of research w.r.t. the choice of projection as well as complexity analysis. Thank you.


Review 3

Summary and Contributions: The paper derives several topological and statistical properties of sliced probability divergences, which have been shown to be useful in practice but not well-studied from a theoretical point of view.

Strengths: The properties seem to be carefully considered and derived, and applicable to any base divergence and its sliced version. The results seem to corroborate and explain observed empirical behavior. Several experiments and examples support the results and a new and effective sliced-Sinkhorn divergence is proposed. I'm convinced by the other reviews and the authors' response that this is a good paper.

Weaknesses: I imagine that the results, while largely theoretical, would be interesting and important for a subcommunity of NeurIPS,

Correctness: This is far outside my area so all I can say is that the paper seems carefully and confidently written, with a thorough understanding of its context.

Clarity: The paper is well-written in the sense of grammar, etc. but I imagine that even for an expert it would be dense and terse. It seems like a huge amount of expertise and deep understanding of prior work would be required to fully understand the derivations.

Relation to Prior Work: Yes.

Reproducibility: Yes

Additional Feedback: I apologize that this paper is so far outside my field of expertise that I'm unable to provide a useful review. I hope that other reviewers are more knowledgeable and constructive.


Review 4

Summary and Contributions: The authors investigate the properties of sliced divergences, which have been used in machine learning as a computationally friendly alternative to their unsliced counterparts for high dimensional applications. Technical results are stated including, but not limited to, characterising the slicing operations ability to preserve metric properties and metrize the weak topology, and sample complexity results.

Strengths: The technical results are nontrivial and of significant importance and novel enough on their own. The promising experimental results are a welcome bonus.

Weaknesses: None

Correctness: I am satisfied with the correctness and empirical methodology.

Clarity: The paper is very well written, with clear, concise notation. I think the statement of Theorem 1 would be improved by use of more mathematical notation to break up the text. L#121 Since the push-forward operation is defined right next to the sliced Wasserstein divergence, I think it would make sense to move the definition of the θ* notation here too. It took me a couple minutes to locate the star notation definition in the boilerplate at the beginning of section 2.

Relation to Prior Work: The authors give an extensive and thorough account of previous work and how their results build upon the results in this area.

Reproducibility: Yes

Additional Feedback:

[Author Response · NeurIPS 2020]

We thank the reviewers for their time and valuable feedback. We are happy to see that the four reviews are positive and
describe our paper as well-written, clear, technically correct and interesting. We provide detailed responses below.

**Reviewer #2:** We thank the reviewer for giving a positive evaluation and valuable comments.

*"Could you elaborate how the Lipschitz assumption can be relaxed?"* The Lipschitz assumption is only used for bounding
the integrand in (S28) in the supplementary document, as demonstrated in the derivations below line 180 (supp. doc).
By assuming the uniform continuity instead, we can adapt these derivations to obtain another bound, but this requires
the introduction of additional assumptions on the modulus of continuity. In that sense, we acknowledge that our remark
might be confusing and decided to remove it from the main text to make the paper more self-contained.

*"One major advantage of the sliced divergences is the dimension-independent sample complexity. But is it meaningful*
*to compare the sample complexity of a sliced divergence and its based divergence?"* This is a fair question and the
reviewer is right in saying that the sliced divergence and its base divergence might have *"different scales"* based on
Corollary 1. The main focus of our paper is to show that slicing results in a dimension-free convergence rate while
carrying out a lot of useful topological properties of the base divergence (e.g. metric axioms, weak convergence). If the
focus is on sustaining such topological properties, then we would argue that the improvement in the convergence rate is
indeed meaningful. On the other hand, slicing also results in less discriminant divergences, as we mentioned for the
IPM case (line 161), and in such a case, the improvement in the rate might be less significant. More analysis is required
to understand the potential reduction in the discriminative power, and we leave it out of scope of this study. We will add
a discussion about this point in the manuscript, and we will also make the suggested changes to the statement in line 53.

*"Proposition 2 can be sharpened using the concentration of measure on the sphere"* We are grateful to the reviewer for
explaining this technique, which indeed leads to an improved bound. We will update Proposition 2 accordingly.

*"Are there clean examples where $S\Delta(\mu_n, \nu_n) \to 0$, but $\Delta(\mu_n, \nu_n) \not\to 0$ for TV distance or for other divergences with*
*unbounded domain?"* We are not aware of such example. We plan on further investigating this non-trivial question as
future work, since this might help understanding whether Theorem 3 can be extended to non-compact domains.

**Reviewer #3:** We thank the reviewer for the positive evaluation and constructive comments. We will include the
suggested additional references in our paper and clarify their connection with our contributions.

*"When comparing distributions, projections onto some directions are arguably more "important" in terms of distinguishing*
*distributions. will this improve the results?"* We agree with the reviewer that sampling the projection directions
uniformly on the sphere is not an optimal choice. However, we underline that this is the most common method used
in practice, and this is why our paper relies on this technique. The study of sliced probability divergences based on
non-uniform distributions is actually a very recent research topic, which raises interesting new challenges [1]. In that
sense, investigating the consequences of the sampling scheme on our results, especially on Theorem 2 and the projection
complexity as suggested by the reviewer, is a great idea: according to our derivations below line 80 in the supplementary
document, Theorem 2 holds for *any* density $\sigma$ defined on the unit sphere ; the bound in Theorem 6 illustrates the effects
of the sampling distribution on the Monte Carlo approximation error, and might help tuning this distribution.

**Reviewer #4:** We thank the reviewer for their positive feedback and evaluation. *"While I imagine the result would be*
*interesting and important for a subcommunity of NeurIPS, IMHO it looks more like it belongs in a math journal"* We
believe that our paper is a good fit for NeurIPS, since our theoretical contributions revolve around metrics that form
the basis of several computational statistics/machine learning methods. Understanding the performance of practical
algorithms by providing theoretical guarantees is an important task in machine learning. In particular, the level of
technicality of our work is very similar to the recent literature on similar topics [2, 3, 4], which were all published at
NeurIPS 2019 and AISTATS 2019.

**Reviewer #5:** We thank the reviewer for their highly positive evaluation. As suggested, in order to increase clarity,
we will revise the statement of Theorem 1 and move the explanation for the notation $\theta^*$ next to the definition of the
Sliced-Wasserstein distance.

## References

[1] Khai Nguyen, Nhat Ho, Tung Pham, and Hung Bui. Distributional Sliced-Wasserstein and applications to generative
modeling, 2020.
[2] Aude Genevay, Lénaïc Chizat, Francis Bach, Marco Cuturi, and Gabriel Peyré. Sample complexity of Sinkhorn
divergences. In *Proceedings of Machine Learning Research*, volume 89. 2019.
[3] Kimia Nadjahi, Alain Durmus, Umut Simsekli, and Roland Badeau. Asymptotic guarantees for learning generative
models with the Sliced-Wasserstein distance. In *Advances in Neural Information Processing Systems 32*. 2019.
[4] Gonzalo Mena and Jonathan Niles-Weed. Statistical bounds for entropic optimal transport: sample complexity and
the central limit theorem. In *Advances in Neural Information Processing Systems 32*. 2019.


[Meta-Review · NeurIPS 2020]

Reviewers agreed that this is a strong paper with a clear technical content, and the rebuttal+discussion contributed even more to raise the opinion on its content. Congratulations to the authors for a very nice contribution.